# Cell-to-cell heterogeneity in Sox2 and Bra expression guides progenitor motility and destiny

**Michèle Romanos[1,2†], Guillaume Allio[1†], Myriam Roussigné[1], Léa Combres[1], Nathalie Escalas[1], Cathy Soula[1], François Médevielle[1], Benjamin Steventon[3], Ariane Trescases[2], Bertrand Bénazéraf[1*]**

[1]Molecular, Cellular and Developmenrtal biology department (MCD), Centre de Biologie Intégrative (CBI), Université de Toulouse, CNRS, UPS, Toulouse, France; [2]Institut de Mathématiques de Toulouse UMR 5219, Université de Toulouse, Toulouse, France; [3]Department of Genetics, University of Cambridge, Cambridge, United Kingdom

**\*For correspondence:**
bertrand.benazeraf@univ-tlse3.fr

[†]These authors contributed equally to this work

**Competing interests:** The authors declare that no competing interests exist.

**Abstract** Although cell-to-cell heterogeneity in gene and protein expression within cell populations has been widely documented, we know little about its biological functions. By studying progenitors of the posterior region of bird embryos, we found that expression levels of transcription factors Sox2 and Bra, respectively involved in neural tube (NT) and mesoderm specification, display a high degree of cell-to-cell heterogeneity. By combining forced expression and downregulation approaches with time-lapse imaging, we demonstrate that Sox2-to-Bra ratio guides progenitor's motility and their ability to stay in or exit the progenitor zone to integrate neural or mesodermal tissues. Indeed, high Bra levels confer high motility that pushes cells to join the paraxial mesoderm, while high levels of Sox2 tend to inhibit cell movement forcing cells to integrate the NT. Mathematical modeling captures the importance of cell motility regulation in this process and further suggests that randomness in Sox2/Bra cell-to-cell distribution favors cell rearrangements and tissue shape conservation.

## Introduction

Cells are the functional units of living organisms. During embryogenesis, they divide and specify in multiple cell types that organize spatially into tissues and organs. Specification events take place under the influences of the cell's own history and of environmental clues. Over the last years, access to new technologies has revealed that embryonic cells often display an unappreciated level of heterogeneity. For instance, gene expression analyses suggest that, within the same embryonic tissue, cells that were thought to be either equivalent or different are actually organized into a continuum of various specification states (*Farrell et al., 2018*; *Wagner et al., 2018*). The impact of this new level of complexity on morphogenesis has not been extensively explored due to the difficulty of experimentally manipulating expression levels within targeted populations of cells in vivo. Progenitor cells located at the posterior tip of the vertebrate embryo, in an area known as the progenitor zone (PZ), constitute a great model to study how a population of stem-like cells develops into different cell types. The use of fluorescent tracers in bird and mouse embryos has revealed that cells of the PZ, called here posterior progenitors, contribute to formation of the presomitic mesoderm (PSM), the mesodermal tissue that generates muscle and vertebrae but also of the neural tube (NT), the neuro-ectodermal tissue that gives rise to the central nervous system (*Selleck and Stern, 1991*; *Iimura et al., 2007*; *Wilson and Beddington, 1996*; *Psychoyos and Stern, 1996*). These studies also evidenced different cell behaviors with some cells exiting the PZ and others remaining resident

in this area. Grafting experiments next showed that resident posterior progenitors have the capacity to self-renew while providing new neural and mesodermal progenies (*Cambray and Wilson, 2002*; *McGrew et al., 2008*), thus indicating that the PZ contains progenitors of different tissues. Heterogeneity in the progeny of PZ cells was further confirmed by retrospective clonal analysis studies performed in the mouse embryo which revealed the existence of single progenitors, giving rise either to neural or mesodermal cells, but also of bi-potent progenitors, named neuro-mesodermal progenitors (NMPs), that generate both neural and mesodermal cells (*Tzouanacou et al., 2009*). The existence of bi-potent progenitors has since been shown at earlier stages of zebrafish development (*Attardi et al., 2018*) and in bird embryos (*Solovieva et al., 2020*; *Wood et al., 2019*; *Guillot et al., 2021*) (for reviews *Wymeersch et al., 2021*; *Sambasivan and Steventon, 2020*). Thus, to sustain the formation of tissues that compose the vertebrate body axis, the heterogeneous population of posterior progenitors must maintain an appropriate balance between the two choices of staying in place and self-renew or exit the progenitor region to contribute to the formation of the NT and the PSM. How this balance is established and controlled over time remains an open question.

Two transcription factors, Sox2 (SRY sex-determining region Y-box 2) and Bra (Brachyury), have been described for their respective roles in neural and mesodermal specification during embryonic development (*Herrmann et al., 1990*; *Bergsland et al., 2011*). Sox2 is known to be expressed in the neural progenitors that form the NT where it contributes to maintain their undifferentiated state. Its involvement in the neural specification has also been revealed by a study showing that ectopic expression of Sox2 in cells of the PSM is sufficient to reprogram these cells, which then adopt a neural identity (*Takemoto et al., 2011*). Bra protein was initially identified for its essential function in the formation of the paraxial mesoderm during the posterior extension phase (*Herrmann et al., 1990*; *Wilson et al., 1995*). Its crucial role in mesodermal specification has been demonstrated, in particular, by phenotypic study of chimeric mouse embryos composed of both Bra mutant and wild-type cells, and in which only wild-type cells are capable of generating posterior mesoderm (*Wilson and Beddington, 1997*). More recent studies have shown that Sox2 and Bra are expressed in posterior progenitors of developing embryos, indicating that activation of their expression takes place in progenitor cells before these cells colonize the NT or the PSM (*Olivera-Martinez et al., 2012*; *Wymeersch et al., 2016*; *Martin and Kimelman, 2012*). In addition, these studies have shown that both proteins are co-expressed in progenitor cells, an observation consistent with the presence of bi-potent progenitors in this tissue. Importantly, it has been found that mouse progenitors display regional differences in Sox2 and Bra expression levels with regions of higher Sox2 in the PZ being neural-fated, and those with higher Bra being mesoderm-fated (*Wymeersch et al., 2016*). Works done in mouse embryos and in in vitro systems derived from embryonic stem cells indicate that Bra and Sox2 influence the choice between neural and mesodermal lineages by their antagonistic activities on the regulation of neural and mesodermal gene expression (*Koch et al., 2017*; *Veenvliet et al., 2020*).

In this study, we aimed at understanding further the relationships between the processes of cell specification and tissue morphogenesis within the PZ, with a particular attention to cellular mechanisms underlying the tightly regulated balance between maintenance of residing posterior progenitors and production of exiting cells that contribute to the formation of mesodermal and neural tissues. By analyzing Sox2 and Bra expression in the PZ of the quail embryo, we show that these proteins are expressed with various levels from one cell to another, thus highlighting an important degree of cell-to-cell heterogeneity in this area. Using overexpression and downregulation approaches, we provide evidence that the relative levels of Sox2 and Bra proteins are a key determinant for posterior progenitor choice to stay in place or exit the PZ to join their destination tissues (neural and mesodermal). Time-lapse experiments further revealed that most posterior progenitors are highly migratory without strong directionality. Functional experiments then revealed that heterogeneous levels of Sox2 and Bra control cell motility: Bra promotes cell motility whereas Sox2 inhibits it indicating a crucial role of motility in guiding progenitors segregation. Mathematical modeling of this process suggests that the spatial distribution of Sox2/Bra heterogeneity is an important factor regulating morphogenesis. Indeed, while graded expression of Sox2/Bra confers a higher short-term stability in PZ shape, random distribution provides a higher rate of elongation, tissue fluidity, and long-term conservation of tissue shape.

## Results

### Levels of Sox2 and Bra proteins display high spatial cell-to-cell variability in the PZ

The transcription factors Sox2 and Bra are known to be co-expressed in progenitors of the PZ (*Oli-vera-Martinez et al., 2012*; *Wymeersch et al., 2016*). As they differentiate from posterior progenitors, neural cells maintain Sox2 expression and downregulate Bra while mesodermal cells downregulate Sox2 and maintain Bra expression. Although Sox2 and Bra are recognized to be key players in driving neural and mesodermal cell fates, the spatial and temporal dynamics of these events remain to be elucidated. As a first step to address this question, we carefully examined the expression levels of the two proteins in the PZ of the quail embryo at stages HH10–11. As expected, analyses of immunodetection experiments revealed co-expression of Sox2 and Bra in nuclei of all PZ cells (*Figure 1A–C*) (n=8 embryos). Noticeably, we observed a high heterogeneity in the relative levels of Sox2 and Bra proteins between neighboring PZ cells. We indeed found intermingled cells displaying high Sox2 (Sox2$^{high}$) and low Bra (Bra$^{low}$) levels and, conversely, Bra$^{high}$ and Sox2$^{low}$ levels as well as cells in which both proteins appear to be at equivalent levels. This cellular heterogeneity was very apparent when compared to the adjacent nascent tissues, that is, the NT and the PSM, where Sox2 and Bra protein levels were found to be very homogenous between neighboring cells (*Figure 1D–F*). Cell-to-cell heterogeneity in posterior progenitor was detected as early as stages HH5–6, a stage corresponding to initial activation of Sox2 and Bra co-expression in the quail embryo (*Figure 1—figure supplement 1*). We also observed heterogeneous levels of Sox2 and Bra proteins in PZ cells of chicken embryo, indicating that it is not a specific feature of quail (*Figure 1—figure supplement 2*). To infer how Sox2 and Bra protein levels go from being co-expressed in a heterogeneous manner in the PZ to being expressed homogeneously in the nascent tissues, we analyzed variations of their respective levels in a series of seven volumes (containing around 100 cells in each volume) located in a posterior to anterior path (from the PZ to the maturing tissues), corresponding to putative trajectories of PSM or NT cells (*Figure 1G–G′*). Data showed that the average expression level of Sox2 increases (+2.22 folds, n=7 embryos) while that of Bra decreases (−3.81 folds) following the neural path (*Figure 1G*). On the contrary, along the paraxial mesoderm path, the average expression level of Sox2 decreases (−2.12 folds, n=7 embryos) while Bra level first increases in the posterior PSM (1.14 folds, positions 1–2) and decreases anteriorly (−5.06 folds, positions 2–7) (*Figure 1G′*). Next, to define whether the cellular heterogeneity found in the PZ depends more on variability of one of the two transcription factors, we quantified protein levels per nuclei of cells populating the PZ. By plotting Sox2 and Bra levels in individual cells, we noticed a broader distribution for Sox2 levels (coefficient of variation of 41.8%) compared to Bra levels (coefficient of variation of 30.75%) (*Figure 1H*), indicating that the cell-to-cell heterogeneity in the PZ is preferentially driven by differences in Sox2 levels. To quantify Sox2 and Bra heterogeneity, we calculated the Sox2-to-Bra ratio (Sox2/Bra) for each cell of the PZ as well as for cells of the NT and PSM, and compared these values. Our data showed high divergences between the three tissues and confirmed the high heterogeneity previously observed in PZ cells (*Figure 1I*). It must however be noticed that these quantitative data revealed a broad range of cell distribution, highlighting, in particular, the presence of cells in the PZ displaying similar Sox2/Bra values as mesodermal or neural cells. We next asked whether the cellular heterogeneity caused by differences in Sox2 and Bra levels is present in the whole volume of the PZ or displays regionalization in this tissue. To address this issue, we analyzed spatial distribution of the Sox2/Bra values on optical transverse sections performed at anterior, mid, and posterior positions of the PZ (*Figure 1J–J′′′*). This analysis confirmed the heterogeneity of Sox2/Bra values which are equally represented in the mid area of the PZ (*Figure 1J′′*). Cells with a high ratio level (Sox2$^{High}$ Bra$^{Low}$) were found to be more represented in the most dorso-anterior part of the PZ (*Figure 1J′*) and cells with a low ratio level (Bra$^{High}$ Sox2$^{Low}$) were found to be more represented in the most posterior part of the PZ (*Figure 1J′′′*). This particular antero-posterior distribution was further confirmed by tissue expression analysis (*Figure 1—figure supplement 3*). However, it should be noted that variations of Sox2/Bra values were noticed in all these areas, indicating that the Sox2/Bra-related cell-to-cell heterogeneity is present in the whole PZ.

Altogether, our data, highlighting significant variability in Sox2 and Bra protein levels within neighboring progenitors of the PZ, evidence an extensive cell-to-cell heterogeneity of this cell population. Noticeably, despite an overall enrichment of Sox2$^{high}$ cells in the dorsal-anterior part of the

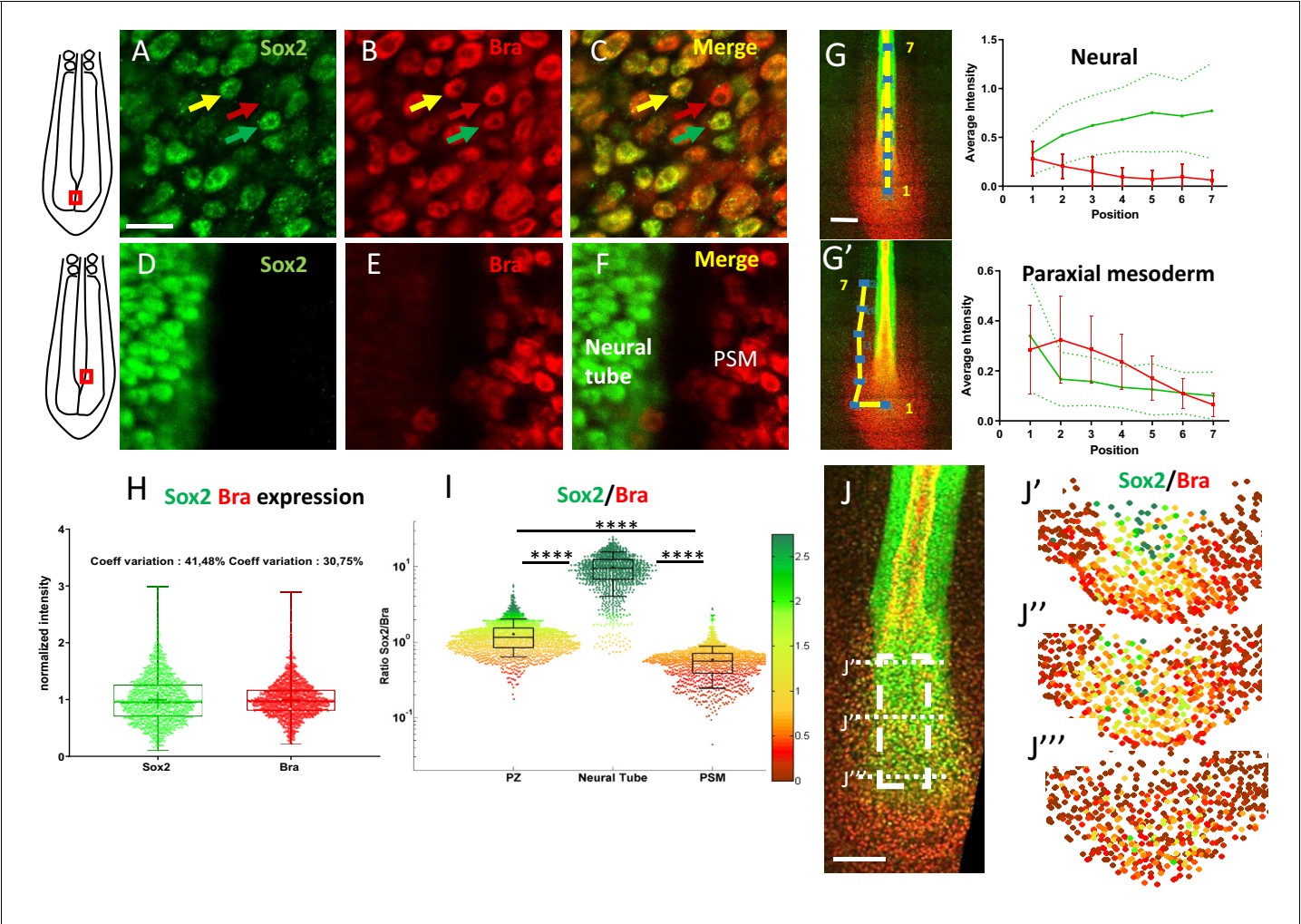

**Figure 1.** Posterior progenitors co-express Sox2 and Bra with a high degree of cell-to-cell heterogeneity. (A–F) Immunodetection of Sox2 (green) and Bra (red) analyzed at the cellular scale in the caudal part of stage HH11 quail embryo, either in the PZ (A–C) or in the nascent NT and the PSM (D–F). Overlay images are presented in (C) and (F). Note cell-to-cell heterogeneity in Sox2 and Bra levels in the PZ, with neighboring cells expressing higher level of Bra (red arrow), higher level of Sox2 (green arrow), or comparable levels of both proteins (yellow arrow), a feature not apparent in the nascent NT and PSM tissues. (G, G') Measurements of Sox2 and Bra levels along putative trajectories (yellow lines) of NT (G) and PSM (G') cells. Fluorescence measurements (blue squares, left images), numbered from 1 to 7, red bars and green dashed lines are errors bars (variability between embryos). (H) Distribution of normalized cell-to-cell expression of Sox2 and Bra in the PZ (n=8 embryos). (I) Cell distribution of Sox2/Bra levels in the PZ (n=9 embryos), the NT (n=7 embryos), and the PSM (n=8 embryos); ratios have been color-coded according to a red (higher Bra) to green (higher Sox2) scale shown on the right side. (J–J''') Representation of the Sox2-to-Bra ratio (green to red same as (I)) in digital transversal sections (40 µm) made in the PZ (dashed lines in the double immunodetection image in (J)). Scale bars=10 µm in (A–F), 100 µm in (G) and (J). NT, neural tube; PSM, presomitic mesoderm; PZ, progenitor zone.

The online version of this article includes the following figure supplement(s) for figure 1:

**Figure supplement 1.** Sox2 and Bra protein co-expression and cell-to-cell heterogeneity start around stage HH5 in quail embryos.

**Figure supplement 2.** Chicken PZ cells co-express Sox2 and Bra proteins with a high degree of cell-to-cell heterogeneity.

**Figure supplement 3.** Sox2 and Bra expression patterns in PZ cells follow opposite gradients at the tissue level.

PZ and Bra[high] cells in the most posterior part, no clear spatial regionalization of these cells was detected, indicating that the PZ is composed of a complex mixture of cells displaying variable Sox2/Bra levels. This variability is further lost as cells enter the NT or the PSM.

## Relative levels of Sox2 and Bra in PZ cells influence their future tissue distribution

The fact that cell-to-cell heterogeneity caused by differences in the Sox2 and Bra levels is observed in PZ cells but not in the PSM and the NT cells was suggestive of a role of these relative protein levels in the decision to leave or not the PZ and to locate in a specific tissue. To test this possibility, we developed functional experiments aimed at increasing or decreasing Sox2 and Bra levels in PZ cells. In the early bird embryo (stages HH4–7), the future posterior progenitors are initially located in anterior epithelial structures: the epiblast and the primitive streak. We thus performed targeted electroporation of progenitors in the anterior primitive streak/epiblast of stage HH5 embryos to transfect expression vectors or morpholinos and further analyzed the subsequent distribution of targeted

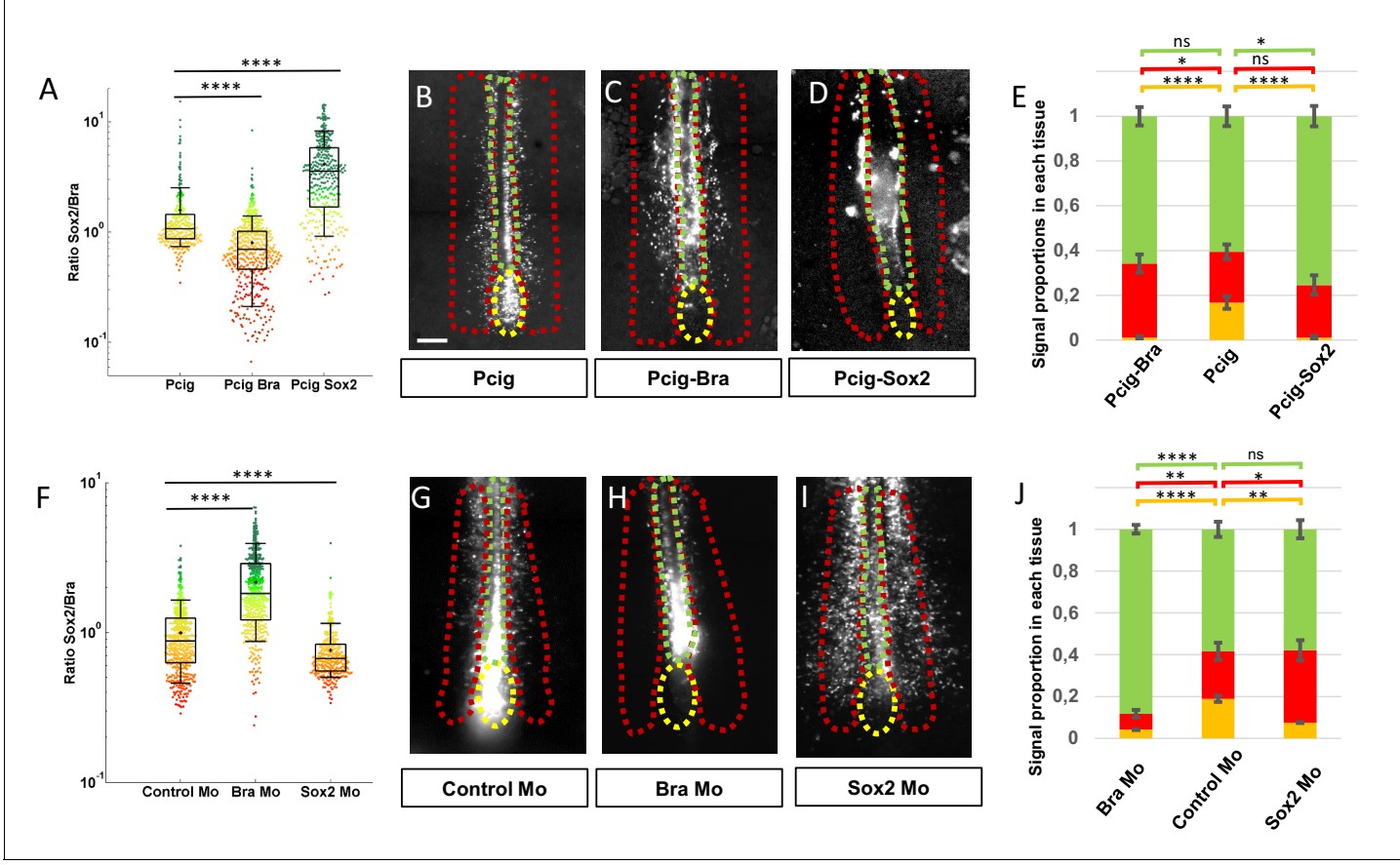

**Figure 2.** Sox2 and Bra levels are critical for progenitor maintenance and tissue distribution. (A, F) Sox2-to-Bra ratios calculated following Bra and Sox2 double immunodetection in the PZ performed 7 hr after electroporation. Values were normalized to the average ratio of non-transfected cells of the same region. (A) Sox2/Bra values in cells transfected with Bra (Pcig-Bra) and Sox2 (Pcig-Sox2) expression vectors compared to cells transfected with the empty vector (Pcig). (F) Sox2/Bra values in cells transfected with morpholinos directed against Bra (Bra-Mo) or Sox2 (Sox2- Mo) compared to cells transfected with a Control-Mo. Ratios were calculated on the basis of 286–590 cells and 3–5 embryos per condition. (B–D), (G–I) Ventral views of embryos collected 20 hr after electroporation showing the GFP signals (white). The PZ, the PSM, and the NT are delineated by yellow, red, and green dash lines, respectively. Expression vectors or morpholinos used are indicated below each picture. Scale bar=100 µm. (E, J) Staked histograms displaying the proportion of cells in the PZ (yellow), the PSM (red), and the NT (green). For each experimental condition, proportion of cells in a given tissue was compared to the same tissue of control embryos by unpaired Student's test (n=27 embryos for Pcig-Bra, n=21 embryos control for Pcig, and n=23 embryos for Pcig-Sox2; n=28 embryos for Bra-Mo, n=27 embryos for Control-Mo, and n=28 embryos for Sox2-Mo). Error bars represent the SEM. NT, neural tube; PSM, presomitic mesoderm; PZ, progenitor zone.

The online version of this article includes the following figure supplement(s) for figure 2:

**Figure supplement 1.** Efficient deregulation of Sox2 and Bra following electroporation of expression vectors and morpholinos.

**Figure supplement 2.** Quantification of Sox2 and Bra protein levels following electroporation of expression vectors and morpholinos.

**Figure supplement 3.** Effect of Sox2 and Bra overexpression on apoptosis and proliferation.

**Figure supplement 4.** Tissue localization of transfected cells after Sox2 and Bra overexpression and downregulation.

cells, focusing on the PZ, the PSM, and the NT (*Figure 2*). As early as 7 hr after electroporation, we could detect the expected modifications of Sox2 or Bra expression in PZ cells for both overexpression and downregulation experiments (*Figure 2—figure supplements 1* and *2*). We observed a significant decrease in the Sox2/Bra levels by either overexpressing Bra or downregulating Sox2 and a significant increase of this ratio when Sox2 was overexpressed or when Bra was downregulated (*Figure 2A,F*). After transfection of expression vectors or morpholinos, we next let the embryos develop until stages HH10–11 and examined fluorescent cell distribution in the different tissues. For this, we measured the fluorescence intensity of the reporter protein (GFP) in the PZ, the PSM, and the NT and calculated the percentage of fluorescence in each tissue. We obtained reproducible data using control expression vector with less than 20% of the fluorescent signal found in the PZ (16.78±2.83%), a little more than 20% in the PSM (22.64±3.30%), and about 60% in the NT (60.57±4.39%) (*Figure 2B,E*). We next found that overexpression of Bra leads to a marked reduction of the fluorescent signal in the PZ (1.17±0.57%) and to an increased signal in the PSM (33.04±4.06%) but has no effect on the NT signal (*Figure 2B,C,E*). Elevating Bra levels is thus sufficient to trigger cell exit from the PZ and to favor integration in the PSM. However, this is not sufficient to impede PZ cell contribution to form the NT. Similarly, we found that overexpression of Sox2 drives exit of the cells from the PZ (1.16±0.67%) favoring their localization in the NT (75.40±4.57%) without significantly affecting proportions of cells in the PSM (*Figure 2B,D,E*). To verify that the differences in fluorescence distributions we observed did not result from distinct apoptotic or proliferation rates, we quantified these parameters 7 hr after electroporation. Our data showed no major changes between the different experimental conditions, validating that protein misregulations indeed act by influencing the distribution of cells in the different tissues (*Figure 2—figure supplement 3*). Spatial distribution of the fluorescent signals obtained using control morpholinos appeared very similar to thoseobserved using the control expression vector (18.93±3.06%, 22.68±4.09%, and 58.38±3.63% for the PZ, the PSM, and the NT, respectively) (*Figure 2G,J*). We found that downregulation of Bra leads to exit of cells from the PZ (4.16±1.57%) and favors cell localization in the NT (88.23±2.04%) at the expense of the PSM (7.59±1.81%) (*Figure 2G,H,J*). Similarly, Sox2 downregulation triggers cell exit from the PZ (7.50±2.35%) and, as expected, leads to higher contribution of cells to the PSM (34.59±4.79%) but this does not occur at the expense of cell contribution to the NT (*Figure 2G,I,J*). These data have been validated by observation of cell distribution on transverse sections (*Figure 2—figure supplement 4*). It must be noticed that, transverse sections showed the presence of a large proportion of Bra-overexpressing cells located in the medial part of the paraxial mesoderm, very close to the NT, while only a few Bra-overexpressing cells were indeed located in the NT. We thus cannot exclude the possibility that, due to this particular cell distribution, the NT signal quantified on whole-mount embryos might have been slightly overestimated (*Figure 2E*). Even if it were the case, it does not question our main conclusions that Bra overexpression, in comparison to control conditions, favors the exit of progenitors from PZ and their subsequent localization into the paraxial mesoderm.

These data, showing that changing the Sox2-to-Bra ratio, tending either toward higher or lower values, is sufficient to trigger cell exit from the PZ, evidence that the relative levels of Sox2 and Bra proteins are the key determinant of PZ cell choice to stay in the PZ or exit this area to enter more mature tissues. Our data also point to the critical influence of the relative levels of Sox2 and Bra in controlling the final destination of cells exiting the PZ, with $Sox2^{high}$ ($Bra^{low}$) cells and $Bra^{high}$ ($Sox2^{low}$) cells preferentially integrating the NT and the PSM, respectively.

## PZ cells are highly motile without strong directionality

To better characterize the movements of posterior progenitors, either staying resident to the PZ or exiting this area, we examined their behaviors using live-cell imaging. We electroporated quail embryos at stage HH5 with a vector encoding for nuclear GFP and performed time-lapse imaging experiments from stage HH8 to stage HH12. At these stages (from stage HH8 onward), posterior progenitors are no longer located in the dorsal epithelium but rather within a dense and internal mesenchymal structure that prefigures the embryonic tailbud (*Schoenwolf and Delongo, 1980* ; *Guillot et al., 2021*). In order to compare migration properties between tissues, we focused on the PZ , the PSM and on the posterior NT (*Figure 3A*, *Figure 3—video 1*). Because the three tissues have a global movement directed posteriorly due to the embryonic elongation, we generated two types of cellular tracking: the raw movement, in which the last-formed somite is set as a reference

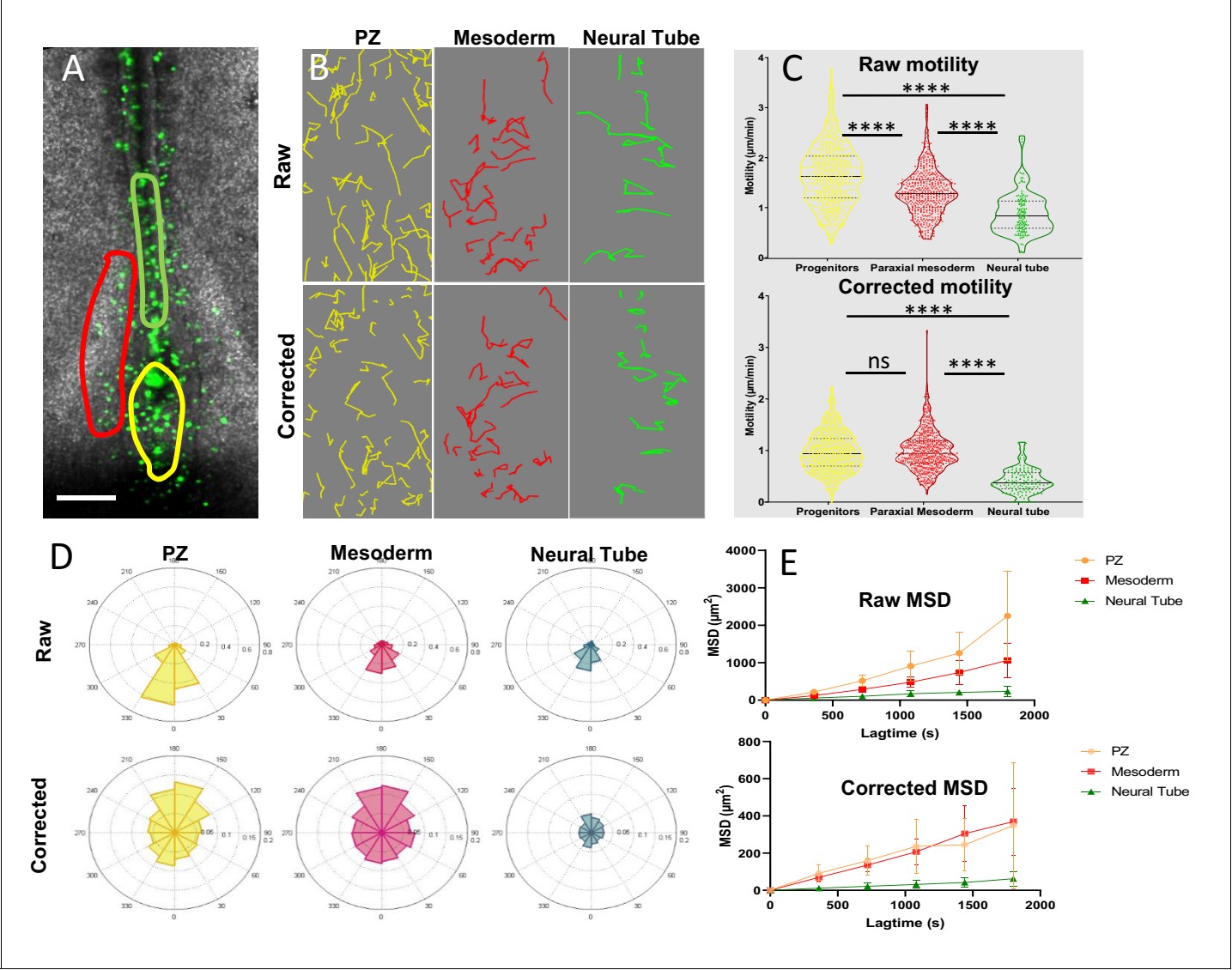

**Figure 3.** Progenitors display high motility without strong directionality. (**A**) Representative image of an H2B-GFP electroporated quail embryo (ventral view) analyzed by live imaging . Transfected cells are detected by the GFP signal (green) . The PZ, the nascent PSM, and the NT are delineated by yellow, red, and green lines respectively. (**B**) Examples of cell trajectories before (raw) and after tissue motion subtraction (corrected). (**C**) Distribution of the raw (top) and corrected (bottom) cell motilities computed in the different regions. (**D**) Directionality of motion assessed by the distribution of angles weighed by the velocity for the different regions, before and after tissue subtraction. (**E**) Assessment of diffusion by analysis of the mean squared displacement in function of time for the different regions (n=7 embryos, 538 cell trajectories analyzed in the PZ, 496 in the PSM, and 128 in the NT). Scale bar=100 µm. NT, neural tube; PSM, presomitic mesoderm; PZ, progenitor zone.

The online version of this article includes the following video and figure supplement(s) for figure 3:

**Figure supplement 1.** Distribution of motility frequencies.

**Figure 3—video 1.** Cellular migration in PZ, NT and PSM (corresponding to *Figure 3*).

https://elifesciences.org/articles/66588#fig3video1

**Figure 3—video 2.** Progenitor migration expressing NLS-Scarlet and Sox2 reporter (N1N2-eGfp-Dest) (corresponding to *Figure 3—figure supplement 1B*).

https://elifesciences.org/articles/66588#fig3video2

**Figure 3—video 3.** Posterior progenitor migration (corresponding to *Figure 3*).

https://elifesciences.org/articles/66588#fig3video3

point, and the 'corrected' movement, in which the cellular movements are analyzed in reference to the ROI (*Figure 3B*). Tracking cell movements allowed for quantification of motility distribution, directionality of migration, and time-averaged mean squared displacement (MSD) (n=7 embryos) (*Figure 3C–E*) . First, we noticed that the average raw motility of PZ cells is higher than that of PSM or NT cells (*Figure 3C*, top panel). Raw directionality was also found more pronounced for PZ cells in the posterior direction compared to PSM or NT cells (*Figure 3D*, upper panel). These results thus confirm that the PZ is moving faster in a posterior direction than surrounding tissues, as previously measured using transgenic quail embryos (*Bénazéraf et al., 2017*). Analysis of local (corrected) motility revealed that PZ cells move in average as fast as PSM cells and significantly faster than NT cells (*Figure 3C*, bottom panel). The distribution of individually corrected PZ cell motilities is however different from the ones of PSM cells as analysis in the PZ showed slower moving cells (PZ corrected motility violin plot in *Figure 3C* is larger for slow values than the PSM counterpart and *Figure 3—figure supplement 1A*), indicating that the motile behavior of PZ cells is more heterogeneous than that of PSM cells. To further characterize the heterogeneity of PZ cell motile behaviors, we co-electroporated a vector coding for a nuclear marker (NLS-Scarlet) with a Sox2 reporter that drives the expression of a destabilized form of eGFP (N1N2-eGFP-Pest). The fluorescence threshold was then adjusted so that only cells emitting high eGFP signal, that is, Sox2high cells, were detected. We then compared motilities of progenitors emitting or not the eGFP fluorescent signal and found that negative cells are more motile than positive cells (*Figure 3—figure supplement 1B*, *Figure 3—video 2*), thus confirming diversity in cell motile behaviors within the PZ and pointing to cells expressing Sox2 as the least motile cells. As previously reported (*Bénazéraf et al., 2010*), we found that, after tissue correction, the motion of PSM cells was mostly non-directional with, however, a slight tendency toward anterior direction which is expected due to the posterior elongation movements of the reference tissue (*Figure 3D*, red plot in the lower panel). The distribution of corrected angles of PZ cell motilities was also found globally non-directed, with however a slight tendency toward anterior direction, to some extent more pronounced than for PSM cells, suggesting that our method is able to detect trajectories of cells exiting the PZ to integrate the NT or the PSM (*Figure 3D*, yellow plot lower panel). Examination of individual cell tracks further confirmed extensive non-directional local migration and neighbor exchanges within the PZ (*Figure 3—video 3*). As PZ cell movement was found being mostly non-directional, we next looked at their diffusive motion by plotting their MSDs, measured in each tissue over time, as it has been previously done for PSM cells (*Bénazéraf et al., 2010*). This analysis showed that the MSD of posterior progenitors is linear after tissue subtraction, as intense as the MSD of PSM cells and significantly higher than that of NT cells, thus demonstrating the diffusive nature of PZ cell movements (*Figure 3E*).

Taken together, these data evidenced that, in the referential of the progenitor region, PZ cell migration is diffusive/without displaying strong directionality (except a slight anterior tendency), with an average motility that is comparable to that of PSM cells and that is significantly higher than that of NT cells. The motility of individual PZ cells is however heterogeneous with some cells exhibiting high motile behavior, as do PSM cells, and others, characterized by higher levels of Sox2 expression, displaying low motility comparable to that of NT cells.

## The Sox2-to-Bra ratio controls motility of PZ cells

To test if Sox2 and Bra could influence progenitor choice of staying in or exiting the PZ and contribute to NT or PSM by controlling cellular motility, we designed experiments combining functional assay and time-lapse imaging in vivo. Sox2 and Bra were either overexpressed or downregulated in PZ cells and the behaviors of posterior progenitors were followed by time-lapse imaging (*Figure 4A–F*). We first monitored raw cell motilities (*Figure 4—figure supplement 1*) and conducted subtraction of the tissue motion to gain insight into local motility and directionality (*Figure 4*). We found that Bra-overexpressing PZ cells display higher motility without significant differences in directionality when compared to control cells. By contrast, when PZ cells overexpress Sox2, we detected a significant reduction of their motility accompanied by an anterior bias in angle distribution compared to control cells (*Figure 4B,C,D*, and *Figure 4—video 1*). We found that Bra downregulation leads to similar significant reduction of cell motility, as well as a change in directionality toward the anterior direction (*Figure 4B,E,F*). Conversely, Sox2 downregulation did not result in significant effect on average cell motility or directionality, even though a tendency toward a slight increase in motility was noticed (*Figure 4B,E,F* and *Figure 4—video 2*). To test if Sox2 and Bra act

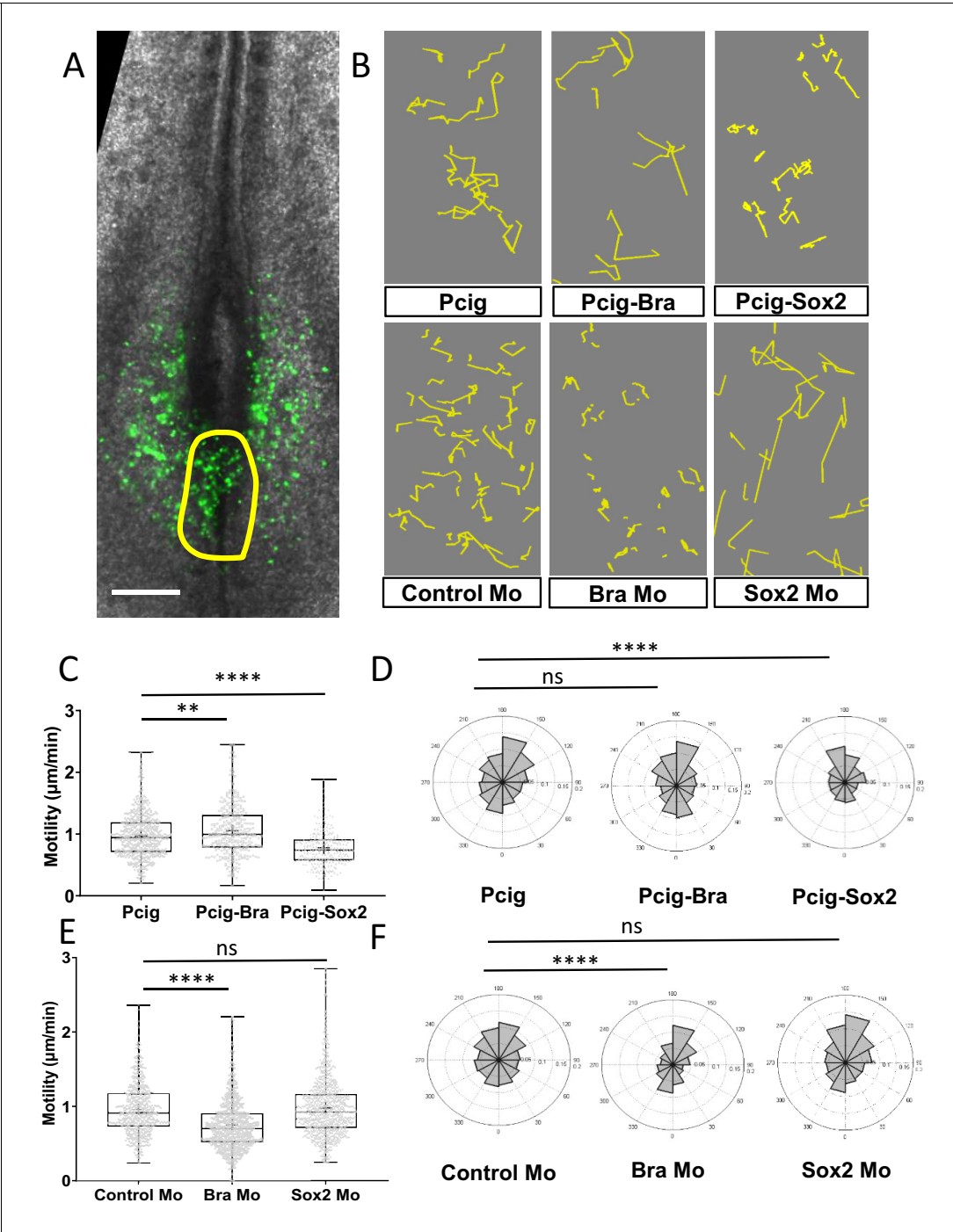

**Figure 4.** Sox2 and Bra deregulations affect progenitor motility. (A) Representative image of a Pcig electroporated quail embryo (ventral view) used to perform progenitor tracking and motility analysis. Transfected cells are detected by the GFP-signal (green) and the PZ is delineated by the yellow line. (B) Examples of cell tracks after correction in embryos electroporated with expression vectors or morpholinos indicated on each panel. (C, E) Distribution of PZ cell motilities after tissue motion subtraction in gain of function (C) and in downregulation (E) experiments. (D, F) Directionality of cell motion after tissue motion subtraction assessed by the distribution of angles in gain of function (D) and in downregulation (F) experiments (n=7 embryos and 541 trajectories for Pcig, n=5 embryos and 307 trajectories for Pcig-Bra, and n=5 embryos and 234 trajectories for Pcig-Sox2; n=5 embryos and 590 trajectories for Control-Mo, n=7 embryos and 753 trajectories for Bra-Mo, and n=5 Embryos and 874 trajectories for Sox2-Mo). Scale bar=100 µm. PZ, progenitor zone.

The online version of this article includes the following video and figure supplement(s) for figure 4:

**Figure supplement 1.** Effects of Sox2 and Bra on raw cell and tissue movements.

**Figure supplement 2.** Cell differentiation in response to Sox2 and Bra overexpression.

*Figure 4 continued on next page*

*Figure 4 continued*

**Figure 4—video 1.** Progenitor migration in Sox2 and Bra overexpression experiments (Corresponding to *Figure 4*).
https://elifesciences.org/articles/66588#fig4video1
**Figure 4—video 2.** Progenitor migration in Sox2 and Bra Mo experiments (Corresponding to *Figure 4*).
https://elifesciences.org/articles/66588#fig4video2

on motility downstream of neural and mesodermal differentiation processes, we checked the differentiation status of progenitors 7 hr after transfection, at a time when progenitors overexpressing Sox2 or Bra have not yet exited the PZ but when effects of Sox2 or Bra misregulation on cell motility can already be measured (data not shown). We found that neither the neural marker Pax6 nor the mesodermal marker Msgn1 was induced in PZ cells overexpressing Sox2 and Bra, respectively (*Figure 4—figure supplement 2*). These data thus strongly support the view that the effects of Sox2 and Bra on PZ cell motility is not a consequence of a drastic change in the differentiation process of progenitor cells.

These data, showing that changing the respective levels of Sox2 and Bra is sufficient to modulate PZ cell motility/migration properties, highlight a key role for these transcription factors in controlling PZ cell movements with Sox2 and Bra inhibiting and promoting cell motility, respectively. When cells have high Sox2/Bra levels, they migrate less and are left behind the PZ to be integrated into the NT. When cells have a low Sox2/Bra ratio, they tend to migrate more, mostly in a diffusive manner, explaining how they leave the PZ to be integrated into the surrounding PSM tissues.

## Modeling spatial cellular heterogeneity and tissue morphogenesis

Our data showed that different levels of Sox2 and Bra affect progenitor motility and regulate their contribution to neural and mesodermal tissues. Cells displaying various levels of these proteins were found intermingled in all regions of the PZ, raising the question of the importance of apparent randomness in their spatial distribution on morphogenesis. Because this question is extremely difficult to tackle experimentally, we turned to agent-based mathematical modeling (*Figure 5*). We set up a model representing developmental times ranging from stage HH8 to stage HH12, a period when the NT, the PSM, and the PZ have already been formed. As there are few dorso-ventral tissue deformations during the selected time window (*Bénazéraf et al., 2017*), we designed a 2D model (X, Y). In this model, PZ cells express dynamic Sox2/Bra levels with a defined probability to switch into a Bra$^{high}$ (Sox2$^{low}$) state (PSM state) or into a Sox2 $^{high}$ (Bra $^{low}$) state (NT state). The motility is directly controlled by the Sox2-to-Bra ratio: Bra $^{high}$ (Sox2$^{low}$) cells display high motility, Sox2$^{high}$ (Bra$^{low}$) cells display low motility, and undetermined progenitors, meaning cells in which the Sox2-to-Bra ratio is still fluctuating, display intermediate motility (*Figure 5A*). Based on the known cell-cell adhesion properties of NT (Sox2$^{high}$) and PSM (Bra$^{high}$) cells, we also considered the Sox2-to-Bra ratio as controlling cellular adhesion so that Sox2$^{high}$ (Bra$^{low}$) cells adhere more to each other than Bra$^{high}$ (Sox2$^{low}$) cells. We as well integrated a non-mixing property between cell types in a way that physical boundaries are maintained between tissues (Appendix 1). Finally, we implemented cell proliferation rates and tissue shape to be as close as possible to biological measurements (*Bénazéraf et al., 2017* Appendix 1). This framework allowed us to model different types of Sox2/Bra spatial distributions within the PZ. We first simulated a distribution that recapitulates the biological data, combining random cell distribution and gradient patterning, that is,. cell-to-cell variations combined with an enrichment in Sox2$^{high}$ cells anteriorly and in Bra$^{high}$ cells posteriorly, as seen in *Figure 1J* (*Figure 5B*). We then verified that this model recapitulates the basic properties of the biological system. Simulations showed that the relative cell numbers (taking into account proliferation) evolve as expected with a stable number of PZ cells and an increased number of NT and PSM cells (*Figure 5C*). We also found that this model reproduces general trends with regard to cell motilities and non-directionality of cell movements (*Figure 5D,E*). We next explored the ability of the model to reproduce maintenance of residing posterior progenitors while the NT and PSM extend toward the posterior pole. Looking at different time points of the simulation process, we indeed observed that the PZ is maintained posteriorly during the elongation process (*Figure 5—video 1*). Our biological results pointed out a critical role of the motility control, however, in our model, Sox2 and Bra control cell motility, adhesion, and non-mixing properties. Thus, we wanted to know if motility is

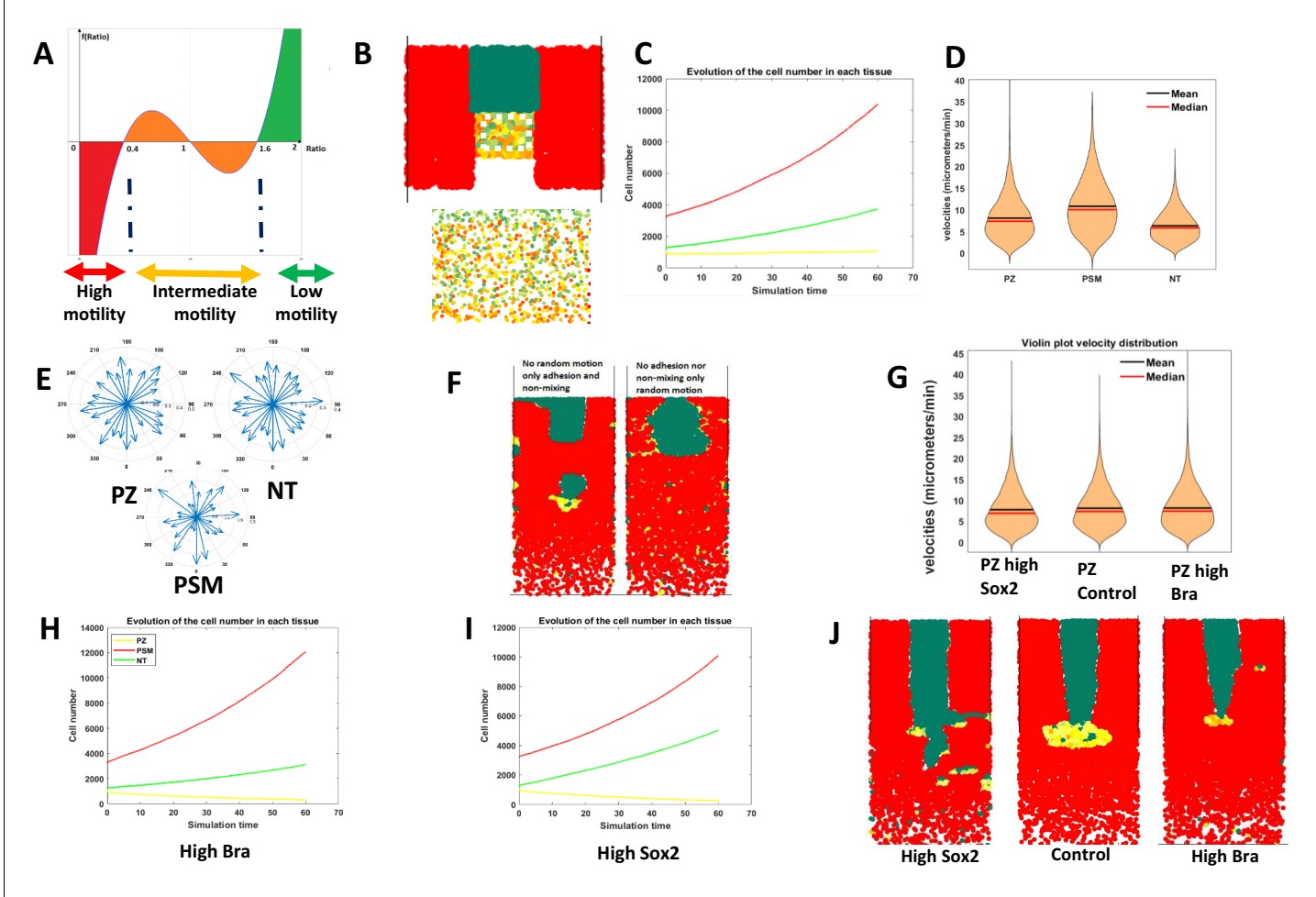

**Figure 5.** Mathematical modeling of progenitor behaviors downstream of Sox2 and Bra heterogeneous expression. (**A**) Graphical representation of the mathematical function defining the Sox2-to-Bra ratio dynamics. The Sox2/Bra value oscillates randomly from 0.4 to 1.6 and noise in the system ensures that some cells pass below 0.4 to be specified into PSM cells (red) while some cells pass above 1.6 to become NT cells (green). Low ratios (below 0.4) confer high motility , high ratios (above 1.6) inhibit motility and ratios between 0.4 and 1.6 confer intermediate levels of motility. (**B**) Posterior region showing the spatial heterogeneity of Sox2/Bra levels with a close-up on the PZ on the bottom panel. (**C**) Evolution of the number of each cell type over time. (**D**) Distribution of cell motilities. (**E**) Directionality of migration in the three tissues. (**F**) Simulation without random motility (left) or without non-mixing and adhesion (right) for progenitors. (**G**) Effects of deregulations of the Sox2/Bra values on cell motility (**G**), cell numbers (**H, I**), and on tissue evolution at 10 hr (**J**). NT, neural tube; PSM, presomitic mesoderm; PZ, progenitor zone.

The online version of this article includes the following video(s) for figure 5:

**Figure 5—video 1.** Mathematical simulations: combined random and graded expression of Sox2 and Bra (mixed).
https://elifesciences.org/articles/66588#fig5video1

**Figure 5—video 2.** Mathematical simulation of Sox2 and Bra overexpression on the mixed model.
https://elifesciences.org/articles/66588#fig5video2

indeed an important parameter in comparison to adhesion and non-mixing properties of progenitors in our model. It turns out that without either random motion or adhesion/non-mixing properties, integrity of the tissues is severely affected (*Figure 5F*), revealing the importance of these parameters in progenitors' behavior in controlling posterior tissue morphogenesis. To challenge further this model, we next tested its ability to recapitulate the experimental results we obtained by overexpressing or downregulating Sox2 and Bra. For this purpose, we explored the consequences on tissues and cell behaviors of numerically deregulating the Sox2/Bra values. As a result, Bra[High] values increase PZ cell motility (*Figure 5G*), lead to generation of a higher number of PSM cells (*Figure 5H*), to a depletion of cells in the PZ, and to a shorter NT (*Figure 5J*; *Figure 5—video 2*). On the opposite, Sox2[High] values lead to reduced PZ cell motility (*Figure 5G*), to a depletion of PZ

cells, to an increased number of NT cells, and to an enlarged NT (*Figure 5I,J*; *Figure 5—video 2*). This model thus recapitulates with success the main biological effects of Sox2 and Bra on progenitor behaviors. To define which particular properties the distribution of heterogeneity, either random or gradient can confer, we created two extreme versions of this model: a first one in which the distribution of Sox2/Bra values in the PZ are fully random (random model) and a second one in which these values are strictly distributed along opposite gradients, that is, a decreasing gradient of Sox2 and an increasing gradient of Bra along the antero-posterior axis (gradient model). We next compared these two models with the initial mixed model (*Figure 6A*). We found that these two extreme cases (random and gradient models) exhibit the main properties regarding PZ maintenance and progenitor distribution than that observed in the mixed model (*Figure 6—video 1*, *Figure 6—figure supplement 1*), suggesting that randomness and gradient features might both be at work in this system. We then analyzed in detail a set of additional parameters and compared these parameters for the different models. We first measured the distance traveled by the PZ over time to define the

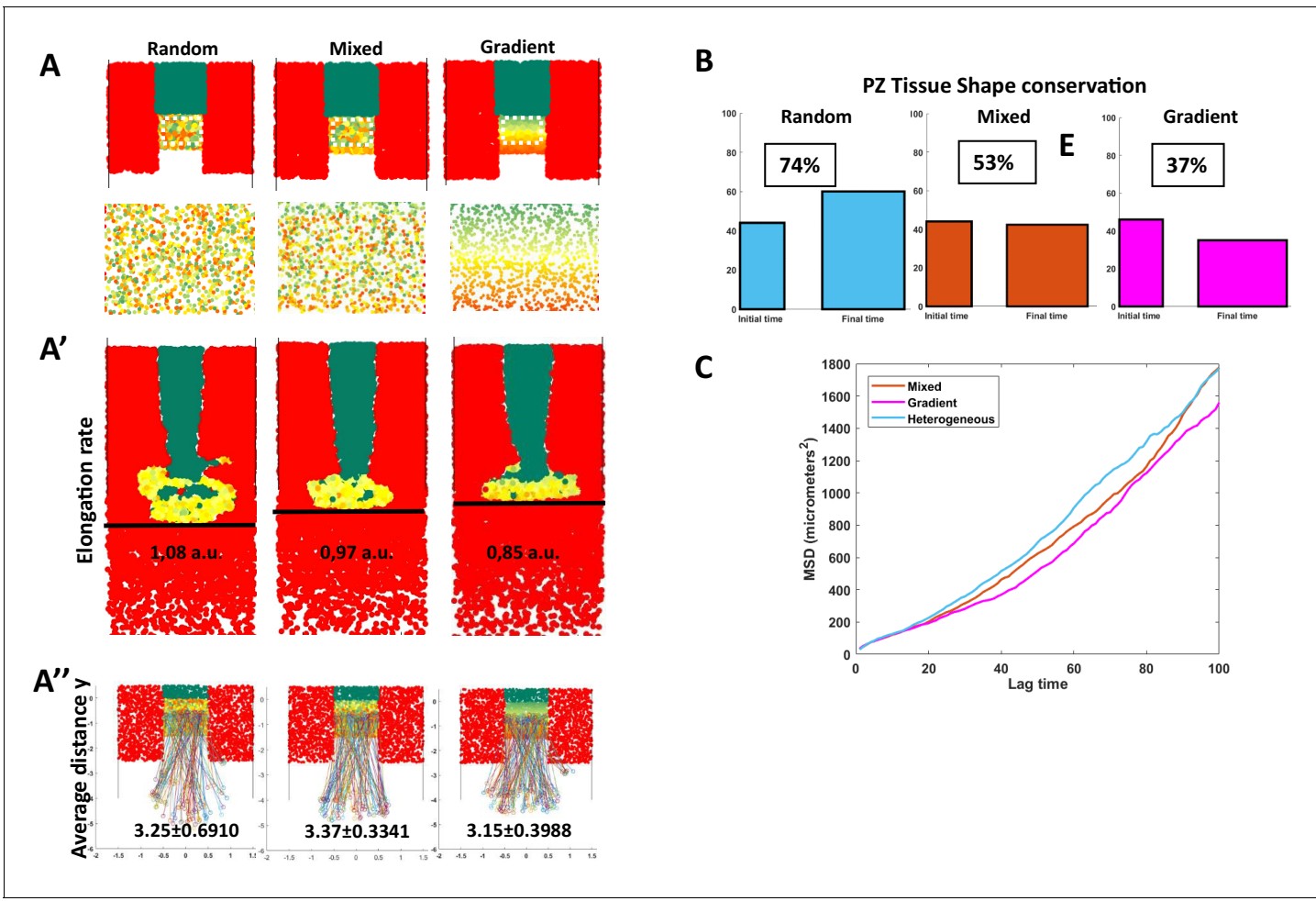

**Figure 6.** Comparison of spatial organizations by modeling. (A) Random model (left), mixed (defined in *Figure 5*), and the graded model following opposite gradients (right). (A) Initial conditions and close-up of the PZ showing the spatial organization of Sox2/Bra levels (bottom panels). (A') Elongation rates measured with the distance traveled by the posterior part of the PZ (black line) in 10 hr. (A'') Y displacement of resident progenitors located at the center (along the anteroposterior axis) of the PZ for each model. (B) Initial (left) and final (right) shapes of the PZ in the different models. Conservation of proportions (length/width) is noted in percentage (100% would correspond to an unchanged shape). (C) MSD calculated for progenitors in the three models. MSD, mean squared displacement; PZ, progenitor zone.

The online version of this article includes the following video and figure supplement(s) for figure 6:

**Figure supplement 1.** Directionality of migration, evolution of cell number, and velocity distribution in the random and gradient models.

**Figure 6—video 1.** Mathematical simulations of random, graded and mixed models.

https://elifesciences.org/articles/66588#fig6video1

elongation speed according to each model. We found differences between the three models, simulation of the random model showed faster elongation than simulation of the gradient model, while simulation of the mixed model resulted in an intermediate speed (*Figure 6A'*). To test if posterior movements of resident progenitors also differ between the three models, we tracked cells that remain in the PZ throughout the simulation and calculated the distances they traveled in the Y direction. This analysis showed that resident progenitors have traveled in a more posterior position in the random and the mixed models than in the gradient model, indicating that random distribution of Sox2/Bra values is more efficient in imposing a posterior movement to these cells than graded distribution (*Figure 6A''*). From the simulation movies, it was obvious that the shape of the PZ was different throughout the three types of simulations (*Figure 6—video 1*). By analyzing the PZ shape at the beginning and at the end of the simulation, we found that proportions (length/width) of the PZ were much more conserved over the elongation process in the random model compared to the gradient model where it became larger (medio-lateral) and shorter (antero-posterior) (*Figure 6B*). Once again, the mixed model gives results that are in between the two extreme models. Finally, to test if the changes we observed at the tissue scale could be due to changes in the diffusivity of cellular migration, we plotted the MSD through time for both the random and the gradient models. We found that the MSD of the random model is higher than that of the mixed model, which itself is higher than the MSD of the gradient model (*Figure 6C*), suggesting that the spatial heterogeneity in the expression of Sox2 and Bra is enhancing the diffusive behavior of PZ cells. This higher diffusivity can therefore bring more tissue fluidity to deform and remodel the PZ and maintain its global shape over long time scales. Indeed, we observed that the PZ, although progressively losing its initial shape in the gradient model, shows fewer transient deformations than in the random model, this stability being inherited in the mixed model (*Figure 6—video 1*).

Taken together, by exploring and comparing Sox2/Bra spatial distributions, our modeling data indicate that while the gradient pattern provides stability by limiting local and transient deformations, random distribution of cell-to-cell heterogeneity could promote cell rearrangements, tissue fluidity, and long-term conservation of tissue shape.

## Discussion

In the present work, we bring evidence that variations of the Sox2-to-Bra ratio in progenitors of the PZ are critical to regulate progenitor motility and tissue destination. Our data support a model in which high levels of Sox2 give cells low motile properties and make them integrate the NT while high levels of Bra rather give them high motile and diffusive properties that push them to exit the PZ and integrate the PSM. Located in-between these low and high motile/diffusive cells are progenitors co-expressing Sox2 and Bra at comparable levels, that are moving with an intermediate speed and remain resident of the PZ as this tissue is moving posteriorly. We propose mathematical models to estimate the importance of spatial cell-to-cell heterogeneity created by variations of the Sox2-to-Bra ratio in the elongation process. As a whole, this study unravels how cellular motility is coupled to progenitors' segregation into different tissues and sheds a new light on how cell-to-cell heterogeneity might ensure robustness in morphogenesis.

In this work, we show that Sox2 and Bra proteins are co-expressed in PZ cells of quail embryos. This co-expression is a conserved property of vertebrate embryos since it has been previously reported in chick, zebrafish, mouse, and human embryos (*Olivera-Martinez et al., 2012*; *Wymeersch et al., 2016*; *Martin and Kimelman, 2012*). Interestingly, it has also been noticed that Sox2 and Bra are expressed at different levels and are therefore heterogeneously expressed in the PZ. In particular, it has been shown that cells from the anterior part of the PZ express high level of Sox2 and are fated toward the NT whereas cells from the posterior part of the PZ are expressing high levels of Bra and are fated toward mesodermal destiny (*Wymeersch et al., 2016*; *Kawachi et al., 2020*). Even though our data showed that such a pattern is apparent in the quail PZ, an important finding of our work is that neighboring cells with variable levels of Sox2 and Bra are found in all areas of the PZ. How the distribution of this particular cell-to-cell heterogeneity is established and further maintained over the elongation process remains an open question. Graded activity of signaling pathways such as Wnt, FGF, and RA, all known to regulate Bra and Sox2 expression (*Ciruna and Rossant, 2001*; *Yamaguchi et al., 1999*; *Gouti et al., 2017*; *Cunningham et al., 2016*;

*Goto et al., 2017*), together with dynamic cross-regulatory activities of Sox2 and Bra (*Koch et al., 2017*) and cell mixing are likely to contribute to create and maintain such a heterogeneity.

The antagonistic interaction between Sox2 and Bra has been proposed to determine fate decision of posterior progenitors (*Koch et al., 2017*). However, this has recently been questioned based on data obtained in mouse showing that Bra does not directly repress Sox2 (*Guibentif et al., 2021*). Interestingly, we also found that Bra-Mo does not lead to an upregulation of Sox2 (*Figure 2—figure supplement 2*). Due to the inhibitory relationships between Sox2 and Bra, it is therefore difficult to know if the ratio between them or the absolute levels are most important to drive their effects. Independently of their regulative interactions, the different levels of Sox2 and Bra expression we observed between posterior progenitors seem indicative of the presence of mixed cell populations in the PZ harboring different specification states: Bra$^{High}$ progenitors being engaged toward the mesodermal fate, Sox2 $^{High}$ toward the neural fate while progenitors with comparable levels of the two proteins being situated in between these two states. In agreement, we found that forced expression of Sox2 and downregulation of Bra favor the integration of posterior progenitors into the NT while forced expression of Bra and downregulation of Sox2 favor their distribution to the PSM. Downregulation or loss of Bra expression has been associated with retention of cells in the progenitor regions in mouse embryo studies, particularly in the tail bud at the level of the CNH (Chordo-Neural Hinge) (*Wilson et al., 1995*; *Wilson and Beddington, 1997*). Although we observed a clear decrease in the number of PZ cells in Bra-Mo compared to control conditions, it is possible that some of the Bra-Mo cells that are remaining in the PZ indeed reside in a region that will contribute to the CNH. Despite clear effects of our experimental approaches on cell localization in the different tissues, results could not explain why, in gain and loss-of-function experiments, preferential distribution of electroporated cells into the NT is not always paralleled by a decrease in their participation to PSM formation (or conversely) (*Figure 2E,J*). A possible explanation is that a progenitor which is already engaged toward a given fate is no longer competent to switch its fate and, thereby, to change its tissue destination. As mentioned above, we found the different types of progenitors intermingled in all areas of the PZ. Single-cell sequencing studies have revealed the molecular signatures of the different progenitor states; however, due to technical limitations, these studies could not reveal their exact locations within the posterior region. Fate maps studies around stages HH4–5 have shown that the distribution of progenitors along the antero-posterior axis of the epiblast/streak is translated in the distribution of their descendants along the medio-lateral axis in formed tissues of older embryos (NT, PSM, and lateral plate) (*Iimura et al., 2007*; *Psychoyos and Stern, 1996*). In this perspective, anterior cells which are expressing high levels of Sox2, give rise to neural cells and more posterior cells, expressing high levels of Bra, give rise to PSM (and eventually to lateral mesoderm for cells located even more caudally). The fact that we found Sox2 and Bra heterogeneously expressed within the PZ is rather suggestive of a more complex picture where position in the progenitor region does not systematically prefigure final tissue destination. Following this scenario, neighboring progenitors could give rise to progeny in different tissues, an observation that is consistent with prospective maps of the PZ in which a small number of labeled cells participate in different tissues (*Selleck and Stern, 1991*; *Iimura et al., 2007*; *Wilson and Beddington, 1996*; *Psychoyos and Stern, 1996*).

Analysis of our time-lapse experiments shows that most PZ cells are highly mobile and that this motility is mainly non-directional. Overexpression of Sox2 or downregulation of Bra strongly inhibits cell motility in the PZ leading to an anterior bias in the direction of progenitor movements. At the opposite, overexpression of Bra and, to some extends, downregulation of Sox2, favor a slight increase in PZ cell motility. Similarly, cells located axially in the zebrafish tailbud have been shown to display highly disordered motility, suggesting conservation of the role of high and non-directional progenitor's motility between vertebrate species (*Lawton et al., 2013*; *Das et al., 2017*). The fact that posterior progenitors often exchange neighbors offers an explanation on how the spatial heterogeneity of posterior progenitors is sorted out to form PSM and NT. Indeed, thanks to their highly migratory properties, Bra$^{High}$ cells could make their way to the surrounding PSM by moving in between other cells including Sox2 $^{High}$ cells that are less motile. It has been shown that Brachyury plays a role in cell migration (*Wilson et al., 1995*; *Wilson and Beddington, 1997*; *Turner et al., 2014*; *Wilson et al., 1993*). In particular, mouse cells that have a mutation in the Brachyury gene have lower migration speed than wild-type cells when isolated and cultured, explaining part of the mouse embryonic axis truncation phenotype (*Hashimoto et al., 1987*). Although a role for Sox2 in

the control of progenitor cell migration has, to our knowledge, not previously been reported, recent works have demonstrated that a rise of Sox2 expression promotes the transition of posterior progenitors to NT during chick embryo secondary neurulation (*Kawachi et al., 2020*) and that turning off Sox2 is necessary for NMP to enter the mesoderm in zebrafish embryo (*Kinney et al., 2020*). In addition, it has also been observed by time-lapse analysis that the dorsal zone between the PZ and the NT does not display excessive cell migration but rather local cell intercalations (*Roszko et al., 2007*; *Gonzalez-Gobartt et al., 2021a*). Taken together, these data confirm the hypothesis that Sox2$^{High}$ cells could be laid down as the PZ moves posteriorly. In our experiments, while a clear inhibition of cell motility can be obtained by Bra downregulation and Sox2 overexpression, only a subtle enhancement of cell motility was obtained by downregulating Sox2 and overexpressing Bra. These differences can be explained by the fact that posterior progenitor's Sox2/Bra ratios and motilities are much more similar to ratios and motilities of PSM cells than NT cells (*Figures 1* and *3*). Biasing progenitors with mesenchymal properties toward a neural state is therefore much more likely to give a difference in motility than a change toward another mesodermal state. In line with this explanation is the fact that during the course of axis elongation posterior progenitors undergo an epithelial-mesenchymal transition before reaching their full potential to give rise to progeny in the NT and the PSM (*Guillot et al., 2021*; *Goto et al., 2017*; *Dias et al., 2020*). Therefore, it is likely that even though Sox2/Bra heterogeneity is present since stage HH5, regulation of progenitor destiny by cellular motility is mostly active after stage HH8 when the progenitors have become mesenchymal with cellular properties that are closer to PSM cells. Indeed, we observed that PZ cell motilities were higher if analyzed between stages HH8 and HH12 compared to the earlier stages HH5 and HH8 (data not shown). Interestingly, the posterior global movement of the PZ region seems constant all along those different stages. Several works, performed in bird embryo, have indicated that physical constraints exerted by neighboring tissues, in particular the PSM, promote the posterior movement of the PZ (*Bénazéraf et al., 2017*; *Bénazéraf et al., 2010*; *Xiong et al., 2020*; *Regev et al., 2017*). It is therefore likely that the posterior movement of PZ cells is the result of both local re-arrangements and external forces acting on the whole region. How Sox2 and Bra are regulating local motility is still an open question. One interesting possibility that is taken into account in our mathematical models is that, as it has been demonstrated during NT dorso-ventral patterning (*Tsai et al., 2020*), differential adhesion between progenitors could regulate their segregation.

Based on our simulations, we propose that both a mix between spatially random and graded patterns heterogeneity in Sox2 and Bra expression are able to maintain progenitors caudally and to guide their progeny in the NT and the PSM. However, little is known about the role of spatial cell-to-cell heterogeneity during morphogenesis. Here, we propose that the spatially random pattern allows more posterior movements, cell rearrangements, and tissue fluidity in the PZ. Interestingly, this fluid-like state as well as disordered cellular movements have been described in PSM tissue to be key for zebrafish embryo axis elongation and morphogenesis (*Lawton et al., 2013*; *Das et al., 2017*; *Mongera et al., 2018*). In addition, the more efficient self-correction observed in the random model is also supportive of spatial cell-to-cell heterogeneity in the PZ providing plasticity to the system. Several studies have shown that this particular region of the embryo is able to regenerate after partial ablation (*Joubin and Stern, 1999*; *Yuan and Schoenwolf, 1999*). Spatial cell-to-cell heterogeneity, which allows easier re-organization of remaining cells than graded cell pattern, thus appears to be an enabling factor for self-correction. Moreover, if gradients of Sox2 and Bra are controlled by secreted signals, tissue ablation could be more detrimental to the diffusion of these signals (and re-patterning) than auto-organization of cell-to-cell heterogeneity. Spatial heterogeneity in gene and protein expression is a common trait of living systems and has been observed in many contexts including, early mouse embryos or cancer cells (*Prasetyanti and Medema, 2017*; *Fiorentino et al., 2020*). The link between cellular spatial heterogeneity and the robustness of morphogenetic processes that we describe here can therefore be relevant beyond the scope of developmental biology.

## Materials and methods

### Quail embryos and cultures

Fertilized eggs of quail (*Coturnix japonica*), obtained from commercial sources, were incubated at 38℃ at constant humidity and embryos were harvested at the desired stage of development. The

early development of quail being comparable to chicken, embryonic stages were defined using quail (*Ainsworth et al., 2010*) or chicken embryo development tables *Hamburger and Hamilton, 1951*. Embryos were grown ex ovo using the EC (early chick) technique (*Chapman et al., 2001*) for 6–20 hr at 39°C in a humid atmosphere.

## Expression vectors and morpholinos

cBra full-length cDNA was cloned by PCR using the following primers (5′-ACCATGGGC TCCCCGGAG-3′; 5′-CTACGCAAAGCAGTGCAGGTGC-3′) into Pcig (*Megason and McMahon, 2002*). cSox2 was cloned from Pccags-cSox2 (*Roellig et al., 2017*) using EcoRV/XbaI into Pcig to obtain Pcig-cSox2. N1N2-eGFP-Pest Sox2 gene reporter gene was obtained from Daniela Roellig (*Roellig et al., 2017*; *Uchikawa et al., 2003*). 3xnls-mScarlet was obtained from Addgene (*Chertkova et al., 2017*). Fluorescein-coupled morpholinos (Mo) were synthesized by Gene Tools. The nucleotide sequences of the morpholinos were designed to target the translation initiation site of quail Bra (5′-AAATCCCCCCCCCCCTTCCCCGAG-3′) and Sox2 (5′-GTACATTCAAACTACTTTTGCC TGG-3′) mRNAs. The Mo (5′-CCTCTTACCTCAGTTACAATTTATA-3′) directed against the transcript of β-human globin was used as control.

## Electroporation

We collected stages HH4–6 quail embryos. The solution containing the morpholinos (1 mM) and pCIG empty (1–2 μg/μl) as a carrier or the DNA solution containing expression vectors Pcig, pCIG-Bra, or pCIG-Sox2 (2–5 μg/μl) were microinjected between the vitelline membrane and the epiblast at the anterior region of the primitive streak (*Iimura and Pourquié, 2008*). The electrodes were positioned on either side of the embryo and five pulses of 5.2 V, with a duration of 50 ms, were carried out at a time interval of 200 ms. The embryos were screened for fluorescence and morphology and kept in culture for up to 24 hr. To observe the distribution of fluorescence in electroporated tissues, embryos were cultured overnight and fixed before being mounted, ventral side up. Transversal sections have been made with a cryostat on embryos embedded in gelatine (*Andrieu et al., 2020*).

## Immunodetection, in situ hybridization and proliferation essay

For immunodetection, embryos of stages HH9–11 were fixed for 2 hr at room temperature in formaldehyde 4% in phosphate-buffered saline (PBS). Blocking and permeabilization were achieved by incubating the embryos in a solution containing Triton X-100 (0.5%) and donkey serum (1%) diluted in PBS for 2 hr. The embryos were then incubated with primary antibodies to Sox2 (1/5000, EMD Millipore, ab5603), Bra (1/500, R and D Systems, AF2085), cleaved Caspase3 (D175, 1/100, CST #9661S), or Pax6 (1/200, MBL, # JM-3636R-100) overnight at 4°C under agitation. After washes, the embryos were incubated with secondary antibodies coupled with Alexa Fluor 555, Alexa Fluor 488 (1/1000, Thermo Fisher Scientific), and with DAPI (4′,6-diamidino-2-phenylindole, 1/1000, Thermo Fisher Scientific D1306) overnight at 4°C under agitation. For in situ hybridization, probes for quail Mesogenin were amplified with the following primers 5′-CGGAGCACTCTGTCTGCTTA-3′ and 5′- TCCCTCATGTTCCTCTGTCA-3′. In situ protocol was adapted from *Denkers et al., 2004*. Proliferation rates were assessed by Edu staining (Click-iT EdU Alexa Fluor 647 Imaging Kit, Thermo Fisher Scientific, C10340) with a pulse of 1 hr duration, for details see *Bénazéraf et al., 2017*.

## Image acquisition, processing, and quantification

Image acquisition for immunodetection and in situ hybridization was performed using Zeiss 710 laser and Leica SP8 confocal microscopes (20×, 40×, 63× objectives). Quantification of Sox2 and Bra levels in 3D was made with Fiji or with the spot function (DAPI staining) of Imaris. Immunodetection signals were normalized to DAPI signal to consider loss in intensity due to depth of the tissue. Immunodetection signals and ratios were calculated and plotted using Matlab. Quantification of protein levels in gain and loss of function experiments was performed 7 hr after electroporation by analyzing immunodetection signal levels within GFP positive progenitors and by normalizing to endogenous expressions measured in non-electroporated cells. Fluorescence distribution in tissues was acquired on a wide-field microscope Axio-imager type 2 (Colibri eight multi-diode light source, 10× objective). Images of electroporated embryos were processed with the Zen software that allows

the assembly of the different parts of the mosaic ('Stitch' function) and were then processed with the 'Stack focuser' plugin of the ImageJ software. The different tissues were delineated on ImageJ with the hands-free selection tool and the images were then binarized using the threshold tool. The total fluorescence intensity emitted by cells transfected with the different constructs was measured and the sum of the positive pixels for the different tissues was calculated. The percentage of fluorescence distribution in the different tissues was then calculated. Immunodetection/proliferation data were quantified using Imaris (Bitplane) and ImageJ/Fiji (*Schindelin et al., 2012*) sofwares.

### Live imaging and cell tracking

Live Imaging was done using Zeiss Axio-imager type 2 (10× objective), as previously described (*Gonzalez-Gobartt et al., 2021b*; *Bénazéraf et al., 2010*). Briefly, stages 7–8 electroporated embryos were cultured under the microscope at 38° in humid atmosphere. Two channels (GFP and brightfield), three fields of views, 10 Z levels were imaged every 6 min for each embryo (six embryos per experiment). Images were stitched and pixels in focus were selected using Stack Focuser (ImageJ). X, Y drift was corrected using MultiStackReg adapted from TurboReg (ImageJ) (*Thévenaz et al., 1998*). Image segmentation was done after background correction using Background Subtractor plugin (from the MOSAIC suite in ImageJ) and cell tracking was done using Particle Tracker 2D/3D plugin (ImageJ) (*Sbalzarini and Koumoutsakos, 2005*). A reference point was defined for each frame at the last formed somite using manual tracking. Regions of interest (ROIs) were defined manually and their posterior movement was defined by manual tracking of the tailbud movement. Subtraction of the tissue movement was done by defining the average motions of cells in the region. Violin plots were generated on Prism 8 (Graphpad). MSD and distribution of angles were calculated and plotted with a Matlab routine. Angle distribution was calculated from trajectories weighted with velocities, and plotted as rosewind plot using Matlab.

### Data representation and statistical testing

Data were plotted using Excel (Microsoft), Prism (Graphpad), Matlab (Mathworks), and PlotsOfData (*Postma and Goedhart, 2019*). Kolmorogov-Sminorv test was used to test for differences in angle distributions in *Figure 4D*,F. For all the other comparisons, unpaired Student's test was used, $p<0.05^*$, $p<0.001^{**}$, $p<0.0001^{***}$, $p<0.00001^{****}$, $p>0.05$ non-significant (ns).

### Mathematical modeling

A cell population of 1100 progenitor cells, 1200 neural cells and 3200 PSM cells was initially distributed in their respective areas. Each cell type was endowed with its proliferation rate, that is, 11.49 hrs for progenitor cells, 10.83 hrs for neural cells, and 8.75 hrs for PSM cells. Each cell I was characterized by a given ratio of Sox2/Bra, named $R_I(t)$, with an assigned value from 0 to 2 (depicted as 0–2 in *Figure 5A* to match with biological ratios), and by a 2D position $(x_I(t), y_I(t))$, each of these variables being time-dependent. In the random model an initial Sox2/Bra ratio value from 0.15 to 0.85 was randomly attributed to progenitor cells. At each time step, each cell updates its Sox2/Bra value through a stochastic differential equation, using the function represented in *Figure 5A* (+noise) and then updates its position (x,y), depending on the value of the ratio, by a biased/adapted random motion. Interaction properties between cells such as adhesion, maximum density, and packing were implemented in the bias of the random motion as detailed in Appendix 1. Simulations focused on the posterior body (1 unit=150 μm). Cells movements in the most anterior region were blocked, considering this region, composed of somites and neuroepithelial cells, as a very dense area, and, similarly, cell passage to either side of the PSM was blocked, considering the lateral plate to be a solid structure.

## Acknowledgements

The authors thank Karine Guevorkian and Eric Théveneau for their critical reading of the manuscript. The authors thank rotation students Nils Vigier, Clément Peux and Khady Moussounda Niang as well as Brice Ronsin and Stephanie Bosch and the CBI imaging facility and Marion Aguirrebengoa for help with statistics. The authors thank Fabienne Pituello for hosting them and for her support. The authors also thank Pierre Degond and members of the Pituello, Soula, Theveneau, and Davy

teams for suggestions and stimulating discussions during the project. This work has been funded by ANR (JC) and the ARC foundation grants.

## Additional information

### Funding

| Funder | Grant reference number | Author |
|---|---|---|
| Agence Nationale de la Recherche | MULTIEL ANRJC | Bertrand Benazeraf |
| Fondation ARC pour la Recherche sur le Cancer | PJA 20191209486 | Bertrand Benazeraf |

The funders had no role in study design, data collection and interpretation, or the decision to submit the work for publication.

### Author contributions

Michèle Romanos, Conceptualization, Formal analysis, Validation, Investigation, Visualization, Methodology, Writing - original draft, Writing - review and editing; Guillaume Allio, Data curation, Formal analysis, Investigation, Methodology, Writing - original draft, Writing - review and editing; Myriam Roussigné, Formal analysis, Investigation, Visualization, Methodology; Léa Combres, Data curation, Formal analysis, Investigation; Nathalie Escalas, François Médevielle, Investigation, Methodology; Cathy Soula, Resources, Investigation, Writing - review and editing; Benjamin Steventon, Conceptualization, Investigation, Writing - review and editing; Ariane Trescases, Resources, Formal analysis, Supervision, Investigation, Methodology, Writing - review and editing; Bertrand Bénazéraf, Conceptualization, Resources, Data curation, Formal analysis, Supervision, Funding acquisition, Validation, Investigation, Visualization, Methodology, Writing - original draft, Project administration, Writing - review and editing

### Author ORCIDs

Myriam Roussigné (iD) http://orcid.org/0000-0002-4240-4105
Benjamin Steventon (iD) https://orcid.org/0000-0001-7838-839X
Bertrand Bénazéraf (iD) https://orcid.org/0000-0002-1937-637X

### Decision letter and Author response

Decision letter https://doi.org/10.7554/eLife.66588.sa1
Author response https://doi.org/10.7554/eLife.66588.sa2

## Additional files

### Supplementary files

• Transparent reporting form

### Data availability

Source data files for Figures 1 and 3 (heterogeneity and motility) are provided on OSF.

The following datasets were generated:

| Author(s) | Year | Dataset title | Dataset URL | Database and Identifier |
|---|---|---|---|---|
| Bénazéraf B, Allio G | 2021 | Movies H2B gfp electroporation quail | https://osf.io/up657/ | Open Science Framework, up657 |
| Bénazéraf B, Allio G | 2021 | Sox2 Bra immunostaining quail embryos stage 10 11HH | https://osf.io/mb5vh/ | Open Science Framework, mb5vh |

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

## Appendix 1

### Part 1: image analysis
#### Fixed tissue imaging
##### Sox2 and Bra intensity profile from PZ to NT or PSM (*Figure 1G,G'*)

Cubes containing approximately 100 nuclei are defined with Imaris along a path going from the PZ to the NT in one case, and from the PZ to the PSM in the other case. The cubes are separated by 70 µm distance approximately. The nuclei detection and data collection are done using the same methods as described previously for the immunodetection analysis. For each cube, the average of the nuclei normalized intensities is calculated and then reported to Graphpad to plot the intensity profile along the PZ-NT path or the PZ-PSM path.

##### Immunodetection analysis and calculation of SOX2/Bra level of expression (*Figure 1H–J*)

After the confocal acquisition, the volume acquired is analyzed with Imaris. The acquisition is done with three color channels: green channel is associated with the level of expression of Sox2, red channel is associated with Bra, and blue channel represents the DAPI staining. The ROIs are drawn with the pointer directly on the confocal imaged volume represented on the Imaris viewer. The dimensions of the ROI can be refined by entering the desired value in the dimension value fields. The 'Spot Detection' function of Imaris is used on the blue channel (DAPI staining) to automatically detect fluorescent nuclei with a user-defined radius equal to the actual radius of a nucleus into the ROI. The data of interest corresponding to the spot detected such as their identities, their position in the three dimensions of the space (X, Y, and Z) , the level of intensity in the three channels (green, red, and blue) are given by the 'statistics' tab. The data are collected on a csv table, opened in an Excel results table where the data is organized by spot identity and their dimensions. The Excel table is read with a homemade Matlab routine where the intensities of the Bra (red channel) and Sox2 (green channel) are normalized with the intensity of DAPI (blue channel) to consider the loss of fluorescent signal due to the depth. This operation is done for each spot. Then, the routine calculates the ratio of the intensity of Sox2 on the intensity of Bra for each spot. The data are plotted with the 'scatter' function of Matlab, in a 2D space representing the projection of the spots along one dimension depending on the orientation of the section used for the analysis. The spots are color-coded with their associated ratio value with a color bar going from red (low ratio, i.e., high Bra intensity, low Sox2 intensity) to green (high ratio, i.e., high Sox2 intensity, low Bra intensity). The scatter plots representing the Sox2/Bra ratio distribution are done with a Matlab code using the 'scatter' function and an adapted version of functions 'beeswarm' (*Stevenson, 2020*) for the distribution plot and 'bplot' (Matlab, *Lansey, 2020*) for the whisker plot (10–90% interval). The Sox2 and Bra expression distribution plots and the calculation of the coefficient of variation are done with Graphpad.

##### Immunodetection analysis for Sox2 and Bra 3D intensity profiles *Figure 1—figure supplement 3*

Raw confocal data are exported to ImageJ and automatically processed with a homemade ImageJ macro. First, a Gaussian filter is applied to the stack before using the binning tool giving a pixel with a value calculated from the average of 4 pixels (approximately 6 µm) in the x and y dimensions of the confocal stack. The macro process an average z-projection every four slices (approximately 8 µm) along the confocal stack with the plugin 'Grouped Z Projector.' Then, a ROI is drawn along the antero-posterior axis with a 40–50 µm width. For each pixel row of the ROI (x dimension), the average of pixels value constituting the row is calculated, giving an intensity profile along the length (y dimension) of the ROI. The operation is done for each average-projected slice. Then, an intensity profile is picked up every four average projected slices (approximately 32 µm), recorded in a CSV results table, and read in Matlab to build a 3D plot of the intensity profiles with the plot3 function.

## Live imaging (*Figures 3* and *4*)

### Movie reconstruction

The movie reconstruction has been done with an ImageJ/FIJI macro that automatizes several steps previously described (*Gonzalez-Gobartt et al., 2021b*). The movie reconstructed and processed is fully on-focus, aligned, and ready for the tracking phase.

### Tracking analysis

The principle of the cell tracking method has been described (*Sbalzarini and Koumoutsakos, 2005*). The cells are tracked with Particle 2D/3D plugin in FIJI using this principle, allowing the reconstruction of trajectories with spots detected frame by frame. The interesting data (trajectory identity, coordinates of trajectory spots along the time-lapse movie (X, Y, Z)) are collected in a csv table results which are then read with a Matlab routine. Manual tracking of the last formed somite and the node are performed respectively to set the reference point and to get the displacement of the posterior area. The first steps of the automatized Matlab routine are to open the csv results table, to draw a reference line along the AP axis on the last frame of the movie using the 'imline' function and to draw the ROIs on the first frame of the movie with 'impoly' function. The ROIs will move along the movie accordingly to the manual tracking of the posterior area. Then, the trajectory spots contained in the ROIs are selected for analysis and their coordinates are corrected accordingly to the reference point (last formed somite). Spots are grouped by trajectory identity and calculations and plots are then performed.

In order to track the Sox2 reporters, the general method is the same as previously described. Once the spot detection is done frame by frame for each channel (Sox2 fluorescence channel, Bra fluorescence channel), the mean intensities are measured for each spot detected in both channels within a radius equal to the cell radius. The spots intensities are then classified depending on a user-defined threshold based on the general background value of the Sox2 fluorescence channel. This classification gives two classes: the Sox2+ class, that is, cells with high expression of Sox2, and Sox2− class, that is, cells with lower expression of Sox2. Then, the calculations and the plots are performed for Sox2+ and Sox2− classes.

### Motility and directionality calculation

A displacement vector is computed for each cell, corresponding to the movement of a cell between two frames. A displacement vector forms a segment of a trajectory, and its norm defines the distance traveled by a cell between these two frames (*Appendix 1—figure 1*). This vector is created using the spot's coordinates of a trajectory in the table results, for every time interval along the trajectory. From this vector, motility and directionality are calculated. The motility corresponds to the norm of the displacement vector $\Delta \vec{r}$ of a cell between two frames divided by the time interval between two frames $\Delta t$.

$$\Delta r(t + \Delta t) = \left\| \Delta \vec{r}(t + dt) \right\| = \sqrt{(x(t + \Delta t) - x(t))^2 + (y(t + \Delta t) - y(t))^2}$$

where x and y are the coordinates of the displacement vector.

$$motility(t + \Delta t) = \frac{\Delta r(t + \Delta t)}{\Delta t}$$

**Trajectory**

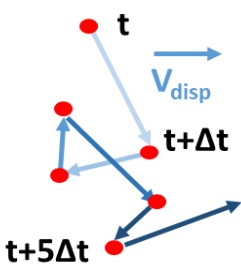

**Displacement/Motility**

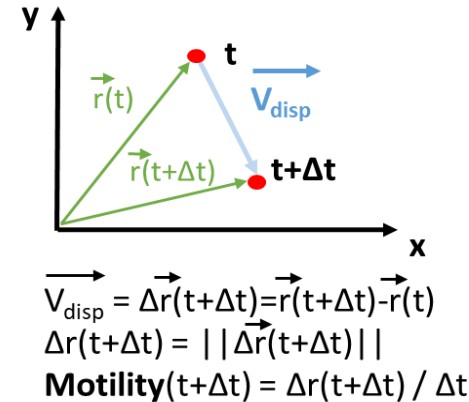

$$\overrightarrow{V_{disp}} = \overrightarrow{\Delta r(t+\Delta t)} = \overrightarrow{r(t+\Delta t)} - \overrightarrow{r(t)}$$
$$\Delta r(t+\Delta t) = ||\overrightarrow{\Delta r(t+\Delta t)}||$$
$$\textbf{Motility}(t+\Delta t) = \Delta r(t+\Delta t) \,/\, \Delta t$$

**Appendix 1—figure 1.** Trajectory and displacement/motility definition and calculation for tracking analysis.

The directionality is the property of being directional or maintaining a direction. In physics, this term is used to describe the preferential sensibility of a sensor or a broadcast system to a direction rather than others. Here, this term is used to describe the directional amplitude of a cell movement, taking into account its motility.

The displacement vector $\left(\overrightarrow{v}\right)$ between two frames is first compared to the reference line drawn previously $\left(\overrightarrow{v_{ref}}\right)$ as shown in **Appendix 1—figure 2**. The angle between the displacement vector and the reference line is calculated for each time interval along the trajectory with the following formula using the cross product and the dot product between the two vectors:

$$\mathrm{Angle(t)} = \tan^{-1} \frac{\left\| \overrightarrow{v}(t) \wedge \overrightarrow{v_{ref}} \right\|}{\overrightarrow{v}(t) \cdot \overrightarrow{v_{ref}}}$$

The angle calculation is done for all the trajectories inside an ROI, and the displacement vectors are then allocated in 12 intervals $(int_\Theta)$ from 0° to 360° (30° intervals) depending on their angle values. The number of vectors allocated to an interval $\left(n\left(v_{disp}(int_\Theta)\right)\right)$ is divided by the total number of displacement vectors inside the ROI $\left(n_{total}\left(v_{disp}\right)\right)$ to get the direction proportion of the displacement vectors $\tau(int_\Theta)$ in the interval.

$$\tau(int_\Theta) = \frac{n\left(v_{disp}(int_\Theta)\right)}{n_{total}\left(v_{disp}\right)}$$

This proportion is weighted by the average motility $<mot\left(v_{disp}(int_\Theta)\right)>>$ calculated from displacement vectors whose angle is contained in this angle interval, giving the directional amplitude $A(int_\Theta)$ of the cell movement in this direction. The calculation is done for every angle interval.

$$A(int_\Theta) = \tau(int_\Theta) \cdot <mot\left(v_{disp}(int_\Theta)\right)>$$

### MSD calculation

The mean square displacement (MSD) measures the deviation of the position of a particle with respect to a reference position over time. The MSD is used to analyze the mode of displacement of a particle and is commonly applied in biophysics for studying cells movements in vivo. Here, the

MSD is calculated for each cell trajectory using the spots coordinates from the table results within a defined time interval $\tau$.

$$MSD(\tau) = <\Delta r^2(\tau) \geq <[r(t+\tau) - r(t)]^2>$$

where $r(t)$ is the position of the particle at time t, and $\tau$ is the lag time between the two positions taken by the particle used to calculate the displacement $\Delta r(\tau) = r(t+\tau) - r(t)$. The MSD is then averaged for all the particles contained in an ROI.

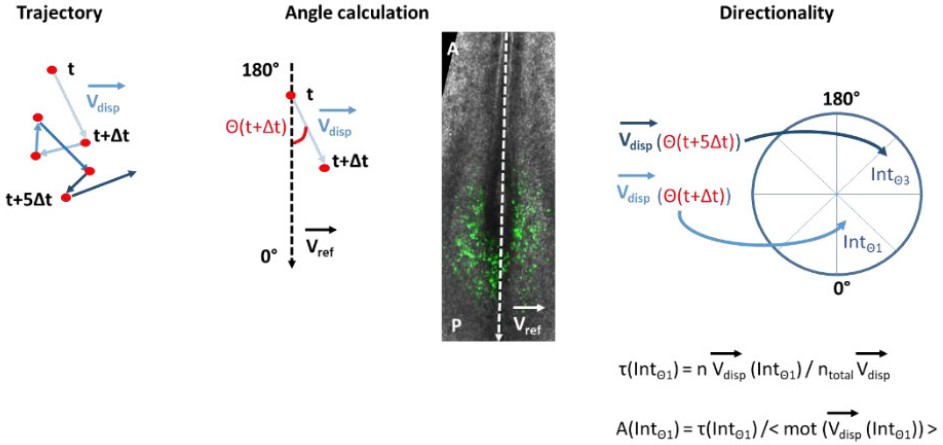

**Appendix 1—figure 2.** Angle and directionality calculation.

## Average vector subtraction

According to *Bénazéraf et al., 2010*, the average movement of electroporated cells in the ROIs corresponds to the extracellular matrix movements. For each ROI, the average velocity vector is defined for each frame by calculating the average velocity and the average direction (i.e., the average angle of the displacement vectors with the reference line) with the coordinates of all the trajectories spots inside the ROI between this frame and the previous one. The average velocity vector is subtracted frame by frame to give the corrected coordinates of all the trajectories inside the ROI, and the corrected motility, MSD and directionality are calculated from these corrected data using the same formulas as previously.

## Save data

The data are collected and are saved in tables in the '.mat' format in a specific folder corresponding to each embryo.

## Plots

The '.mat' files containing the data for plots are read with a Matlab routine which collects the data per variables (motility, MSD, and directionality), conditions, and ROIs for all the embryos. The MSD is averaged for all embryos per lag time and the directionalities are averaged for all embryos per angle intervals determined previously. For the MSD and the motility distributions, table results are opened with Matlab and then transferred to Graphpad. The rose wind plots are made in Matlab according to the number of angle intervals and using the 'rose and polar' functions from Matlab, showing the distribution of cells directionality in angle bins (here, 12 angle bins).

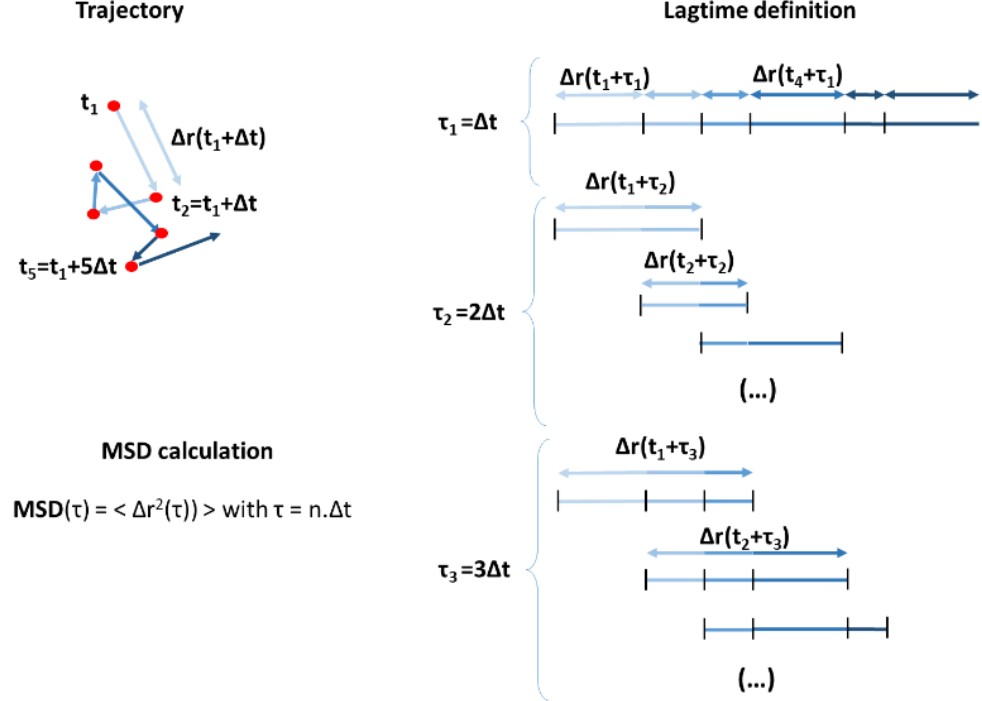

**Appendix 1—figure 3.** Lagtime definition and MSD calculation.

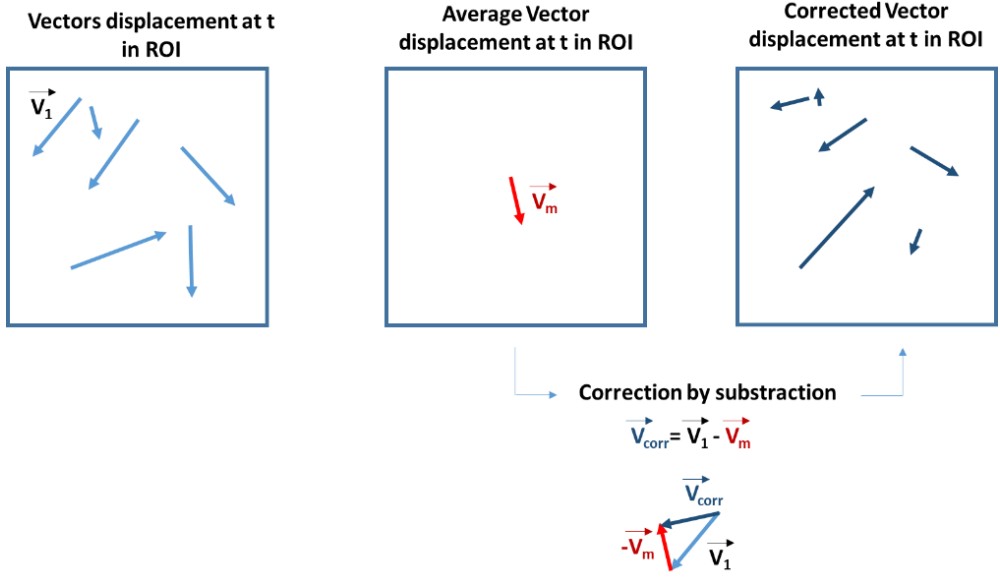

**Appendix 1—figure 4.** Average vector subtraction for vector correction.

## Part 2: mathematical modeling

In this section, we will exhibit the details of the modeling choices. We develop three 2D agent-based models to better understand how the regulation of cell motility by specification factors Sox2 and Bra can influence progenitor behaviors during axis elongation, and to characterize the role of spatial heterogeneity and of a gradient organization of the PZ. We call the three models: (1) the random model

(Sox2/Bra levels are randomly distributed among progenitors), (2) the gradient model (Sox2/Bra levels are patterned in gradient pattern among progenitors), and (3) the mixed model.

## The random model
### Main variables of the model

In this model, we consider a cell population of N cells (with N=5500 at initial condition) the main variables are the cell type (defined by its Sox2/Bra level, details below), which defines its color in the simulation, and the cell position (random motion). Each of these variables is time-dependent.

### Differentiation and evolution of Sox2/Bra levels

Each cell i is characterized by its type, represented here by the ratio $R_i(t) = \frac{Sox2}{Bra+Sox2}$ (see Remark 3): A ratio $R_i(t) = 0$ corresponds to a PSM cell (a red cell in the simulation), and a cell of ratio $R_i(t) = 1$ corresponds to a neural cell (a green cell in the simulation). Finally, a cell with a ratio $0<R_i(t)<1$ is a progenitor cell (different shades of yellow in the simulation), that has not yet differentiated.

To model heterogeneity and oscillations of transcription factors $R_i(t)$ of a cell i we use the following stochastic differential equation:

$$dR_i(t) = \left(100(R_i(t)-0.2)(R_i(t)-0.5)(R_i(t)-0.8)dt + k_r dB_i^R(t)\right)1_{R_i \in (0,1)}$$

with $dB_i^R(t)$ the increment of the Brownian motion associated to the ratio, it represents the various signals a cell receives, and affecting the expression levels of Sox2 and Bra, thus leading it to either cell fate: NT or PSM, corresponding to a ratio of 1 or 0, respectively. Moreover, $k_r$ is a constant representing the intensity of the signal, it is related to the specification rate of the cells. Its value is such that the number of progenitors remains constant throughout the phases of development we are modeling, then in a way to keep the balance between the specification of the cells into either fate and their proliferation (see below for details on the proliferation). Finally, a cell with a ratio $0<R_i(t)<0.2$ is called a pre-mesodermal cell, whereas a cell with a ratio $0.8<R_i(t)<1$ is called a pre-neural cell.

Remark 1: In practice for the numerical simulations, we work discretely in time thus we replace the Brownian motion with a uniform variable $U_i^R(t) \in [-1;1]$ at each time step.

Remark 2: Ignoring the noise, the deterministic function governing the evolution of the ratio has five equilibriums at 0, 0.2, 0.5, 0.8, and 1 with 0, 0.5, and 1 stable equilibriums and 0.2 and 0.8 unstable equilibriums.

Remark 3: The quantity we chose to describe the cell type is the ratio $Sox2/(Bra + Sox2)$. In fact one can have in mind a more general quantity in $[0;1]$ such that 0 and 1 correspond respectively to the PSM and the NT cells, and such that this quantity obeys to the equation of the ratio.

### Diffusion process

From literature (**Bénazéraf et al., 2010**) and the present analysis (**Figure 3**), we know that cells in the PZ, in the PSM, and in the NT have a non-directional motility. Thus, one way to model cell motility is using a random unbiased walk (diffusion process) whose intensity depends on the cell's ratio $R_i$. The stochastic differential equations read:

$$dx_i(t) = k_x dB_i^x(t) \times \mathfrak{V}(R_i(t)),$$

$$dy_i(t) = k_y dB_i^y(t) \times \mathfrak{V}(R_i(t)),$$

with $dB_i^x, dB_i^y$ the increment of the Brownian motion associated to the position (in x and in y) of the cell. They represent the noise of the random walk affecting the cell's movements.

Remark: In practice, for the numerical simulations we work in a discrete environment in time, thus we replace the Brownian motions with uniform variables $v_i^x(t), v_i^y(t) \in [-1;1]$.

Furthermore, the variables $k_x and k_y$ represent the intensity of the diffusion, in accordance with our scale (here we took 1u =150 $\mu m$). Furthermore, the velocity function $\mathfrak{V}(R_i(t))$ has the following form:

$$V(R_i(t)) = \begin{cases} 1 + \beta(\tilde{R} - R_i(t))^2 & \text{if } 0 \leq R_i(t) \leq \tilde{R} \\ 1 & \text{if } \tilde{R} \leq R_i(t) \leq R^* \\ 1 + \alpha(R^* - R_i(t))^2 & \text{if } R^* \leq R_i(t) \leq 1 \end{cases}$$

with $\alpha$ and $\beta$ two positive constants: $\alpha = \frac{1-0.65}{(R^\star-1)^2}, \beta = \frac{1-0.8}{\tilde{R}^2}$, and $R^\star = 0.8$, and $\tilde{R} = 0.2$, are the thresholds that indicte the change of dynamics between progenitors and pre-neural, and progenitors and pre-mesodermal.

## Proliferation

Each cell type is endowed with its average proliferation rate: 11.49 hr for the progenitor cells, 10.83 hr for the neural cells, and 8.75 hr for the PSM cells (*Bénazéraf et al., 2017*).

Then each cell, depending on its type, has a probability to proliferate given by $b_{PSM}$ for PSM cells, $b_{NT}$ for NT cells, and $b_{PZ}$ for progenitors:

$$b_{PSM} = \frac{1}{8.75}, b_{NT} = \frac{1}{10.83}, b_{PZ} = \frac{1}{11.49}.$$

When a mother cell proliferates, it gives rise to one daughter cell, which inherits instantly its position and its ratio. The daughter cell is then integrated in the model and obeys to the equations. Depending on its type (PSM, NT, and PZ), it is added to the total cell number of its corresponding population.

## Biophysical properties and cell-cell interactions

The system is confined between two horizontal lines, at x=1.5 and x=−1.5, representing the lateral plate which, based on its higher cellular density (*Bénazéraf et al., 2017*; *Bénazéraf et al., 2010* ) can be assumed to act as a physical barrier on either side of the PSM. The system is also limited from the most anterior part, at y=2, as we consider everything above that line to be of high density/epithelial (somites and anterior NT). Recall the scale we chose: $1u = 150\mu m$ (u=graph unit), and here we represent a portion of the posterior body (the most posterior), considering that a portion of the PSM and the NT has already been formed (further details in the section Initial condition).

Cells are endowed with properties that take into account possible interactions between cells. Indeed, cells with high Sox2 adhere, more or less depending on their level of Sox2. The adhesion dynamic is to search inside a neighborhood $\epsilon$ (a ball of radius $\epsilon$ representing $15\mu m$) for a pack of cells of the same type (a progenitor). The number of cells in this pack depends on the cell's level of Sox2. Therefore, cells of the NT adhere the most. This is a natural assumption as we know that the NT is an epithelium where cells are closely packed.

Moreover, to guarantee non-mixing, cells tend to move away from foreign high densities within a certain neighborhood (a ball of radius 2ε representing $30\mu m$). In particular, cells with various levels of Bra flee densities of all foreign cells. Furthermore, to keep well-defined physical boundaries between the tissues, and to avoid packing, we also ask neural cells to move away from high densities of PSM cells. Using these interaction rules, we created a model that presents self-organizing phenomena, with physical boundaries between the tissues.

## Initial condition

We consider the embryo to be at a stage where a portion of the NT and the PSM has already been formed (anteriorly), for example at stage HH8. We distribute 1200 neural cells inside the square $[-0.05; 0.5] \times [0; 2]$, with an initial ratio of 1. As these cells are already differentiated into their neural fate, their ratio is not updated at each time step. We distribute 3200 PSM cells, which corresponds to 1600 from either side of the already formed NT, inside $[-1.5; -0.5] \cup [0.5; 1.5] \times [-2.5; 2]$, with an initial ratio of 0. These cells also do not update their ratio as they have already engaged in their mesodermal fate. Finally, we distribute 1100 progenitor cells inside the square $[-0.5; 0.5] \times [-1.5; 0]$.

## Each progenitor cell i is attributed a random initial ratio in [0.15; 0.85]

Remark: By definition of the ratio and by its equation , we can see that progenitor cells with a ratio less than 0.2 are predestined to a mesodermal fate, and cells with a ratio higher than 0.8 are predestined to a neural fate. These pre-destined cells account for 14% of the total number of progenitor cells initially. The presence of these specified cells in our model is justified by the fact that we have noticed the presence of progenitors having Sox/Bra levels as high or as low as NT or PSM cells in the biological system (*Figure 1I*)

List of variables and their values:

$K_r$=0.27,

$\varepsilon$=0.1,

maxTN=5, representing the maximal density (number of cells in a neighborhood of size 2ε) a neural cell can withhold before fleeing the density,

maxPSM=4, representing the maximal density (number of cells in a neighborhood of size 2ε) a PSM cell can withhold before fleeing the density,

maxPZ=4, representing the maximal density (number of cells in a neighborhood of size 2ε) a progenitor cell can withhold before fleeing the density,

$\Delta t = 0.01$, the time step of the scheme, representing 6 s of real biological development,

Tmax=60, representing 10 hr of real biological development,

$K_x$=0.43,

$K_y$=0.43,

$R^\star = 0.8$,

$\widetilde{R} = 0.2$.

Remark: One should note that the frame rate of the live imaging is 1 frame per 6 min. Thus, in our simulations, we save one frame each 6 min (corresponding to $60\Delta t$).

## The gradient model

The purpose of this model is to impose a gradient-like structure to the PZ. More specifically, our aim is to have cells expressing high Sox2 in the most anterior PZ, and cells expressing high Bra in the most posterior PZ, while keeping the pool of progenitors. To do so, we divide the PZ into eight subdivisions in the direction of the y-axis, such that each subdivision contains the same number of cells. Cells in subdivisions 1–3 will be subjected to a signal coding for an overexpression of Sox2, and cells in subdivisions 6–8 will be subjected to a signal coding for an overexpression of Bra, while cells in subdivisions 4 and 5 will be subjected to a weak signal to keep the ratio of these cells near the equilibrium state 0.5, which will allow them to proliferate and keep the progenitor pool. Then, calling $f(R_i(t))100(R_i(t) - 0.2)(R_i(t) - 0.5)(R_i(t) - 0.8)$, the equation for the ratio reads:

$$dR_i(t) = \begin{cases} f(R_i(t))dt + k_r dB_i^{R_a}(t) & \text{if } subdiv(1) \leq y_i(t) \leq subdiv(3) \\ f(R_i(t))dt + k_r dB_i^{R_b}(t) & \text{if } subdiv(4) \leq y_i(t) \leq subdiv(5) \\ f(R_i(t))dt + k_r dB_i^{R_c}(t) & \text{if } subdiv(6) \leq y_i(t) \leq subdiv(8) \end{cases}$$

with $dB_i^{R_a}(t), dB_i^{R_b}(t), dB_i^{R_c}(t)$ the increments of the Brownian motion of the cell in each subdivision, respectively replaced by $U_i^{R_a}(t), U_i^{R_b}(t), U_i^{R_c}(t)$ in the numerical simulations, with $U_i^{R_a}(t)$ a uniform random variable in $[-0.9; 1.1]$ corresponding to a high Sox2 signal in this region, $U_i^{R_c}(t)$ a uniform random variable in $[-1.1; 0.9]$ corresponding to a high Bra signal in this region, and $U_i^{R_b}(t)$ a uniform random variable in $[-0.1; 0.1]$ corresponding to a weak signal in this region where ratios will vary slightly, thus inhibiting specification and favoring the maintenance of the progenitor pool.

Thus, at each time step, cells evaluate their position, and update their ratio accordingly.

Remark: We do not change the velocity function $\mathfrak{V}(R_i)$, nor the interaction between cells nor the biophysical properties of the model. However, the initial distribution of the progenitor cells is changed to be gradient patterned (details in the paragraph Initial condition). The gradient structure is enforced into the new (updated) PZ at each time step.

## Initial condition

The disposition of the tissues PSM, TN, and PZ remains the same as in the random model. The initial distribution of the PSM and NT cells is also unchanged. However, we change the initial distribution

of the PZ cells from a random one, to a gradient distribution as the following: we distribute 1100 progenitor cells inside the square $[-0.5; 0.5] \times [-1.5; 0]$. Each progenitor cell i is attributed an initial ratio in $[0.15; 0.85]$ depending on its position in the PZ: cells with the highest ratio (closer to 1) will be placed anteriorly, whereas cells with the lowest ratio (closer to 0) will be placed posteriorly. Using a linear function in y, the ratios are initially distributed as the following:

$$R_i(t=0) = -\frac{0.15 - 0.85}{1.5} y_i(t=0) + 0.85$$

Remark: By definition of the ratio, its equation, and what we just described, we can see that progenitor cells with a ratio less than 0.2, predestined to a mesodermal fate, are now in the most posterior region of the PZ, and cells with a ratio higher than 0.8, predestined to a neural fate, are now placed in the most anterior part of the PZ.

List of variables and their values:

We kept most variables equal to the ones in the random model, however, some of them are not comparable then we had to change their values to meet the model hypotheses.

$k_r$=0.23

$\epsilon = 0.1$,

maxTN=5, representing the maximal density (number of cells in a neighborhood of size $2\epsilon$) a neural cell can withhold before fleeing the density,

maxPSM=4, representing the maximal density (number of cells in a neighborhood of size $2\varepsilon$) a PSM cell can withhold before fleeing the density,

maxPZ=4, representing the maximal density (number of cells in a neighborhood of size $2\varepsilon$) a progenitor cell can withhold before fleeing the density,

$\Delta t = 0.01$, the time step of the scheme, representing 6 s of real biological development,

Tmax=60, representing 10 hr of real biological development,

$K_x$=0.43,

$K_y$=0.43,

$R^\star = 0.8$,

$\tilde{R} = 0.2$.

## The mixed model

After studying the role of spatial heterogeneity and of a gradient structure of the PZ, we now want to have a model that is close to the biological reality. More specifically, our aim is to have cells expressing high Sox2 in the most anterior PZ, and cells expressing high Bra in the most posterior PZ, while keeping the pool of progenitors in the middle area expressing heterogeneous levels of Sox2 and Bra in a spatially heterogeneous pattern. To do so, we divide the PZ into eight subdivisions in the direction of the y-axis, such that each subdivision contains the same number of cells. Cells in subdivision 1 will be subjected to a signal coding for an overexpression of Sox2 (biased noise), and cells in subdivision 8 will be subjected to a signal coding for an overexpression of Bra (biased noise), while cells in subdivisions 2–7 will be subjected to an unbiased noise, thus creating spatial heterogeneity. Then, calling $f(R_i(t))100(R_i(t) - 0.2)(R_i(t) - 0.5)(R_i(t) - 0.8)$, the equation for the ratio reads:

$$dR_i(t) = \begin{cases} f(R_i(t))dt + k_r dB_i^{R_a}(t) & \text{if } y_i(t) = subdiv(1) \\ f(R_i(t))dt + k_r dB_i^{R_b}(t) & \text{if } subdiv(2) \leq y_i(t) \leq subdiv(7) \\ f(R_i(t))dt + k_r dB_i^{R_c}(t) & \text{if } y_i(t) = subdiv(8) \end{cases}$$

with $dB_i^{R_a}(t), dB_i^{R_b}(t), dB_i^{R_c}(t)$ the increments of the Brownian motion of the cell in each subdivision, respectively replaced by $U_i^{R_a}(t), U_i^{R_b}(t), U_i^{R_c}(t)$ in the numerical simulations, with $U_i^{R_a}(t)$ a uniform random variable in $[-0.9; 1.1]$ corresponding to a high Sox2 signal in this region, $U_i^{R_c}(t)$ a uniform random variable in $[-1.1; 0.9]$ corresponding to a high Bra signal in this region, and $U_i^{R_b}(t)$ a uniform random variable in $[-1; 1]$ corresponding to the unbiased noise creating spatial heterogeneity. Thus, at each time step, cells evaluate their position, and update their ratio accordingly.

Remark: We do not change the velocity function $\mathfrak{B}(R_i)$, nor the interaction between cells nor the biophysical properties of the model. However, the initial distribution of the progenitor cells is changed to be a mixed pattern between the random pattern and the gradient pattern (details in the

paragraph Initial condition). The mixed structure is enforced into the new (updated) PZ at each time step.

## Initial condition

The disposition of the tissues PSM, TN, and PZ remains the same as in the random and in the gradient model. The initial distribution of the PSM and NT cells is also unchanged. However, we change the initial distribution of the PZ cells to a mixed distribution as the following: we distribute 1100 progenitor cells inside the square $[-0.5; 0.5] \times [-1.5; 0]$. Each progenitor cell i is attributed an initial ratio in $[0.15; 0.85]$, depending on its position in the PZ: cells in subdivision 1 (high in Sox2) are attributed a random ratio between 0.5 and 0.85, and cells in subdivision 8 are attributed a random ratio between 0.15 and 0.5. Finally, cells in the remaining subdivisions acquire a random ratio between 0.15 and 0.85.

List of variables and their values:

We kept most variables equal to the ones in the random and gradient models, however, some of them are not comparable then we had to change their values to meet the model hypotheses.

$k_r$=0.24,

$\varepsilon$=0.1,

maxTN=5, representing the maximal density (number of cells in a neighborhood of size $2\varepsilon$) a neural cell can withhold before fleeing the density,

maxPSM=4, representing the maximal density (number of cells in a neighborhood of size $2\varepsilon$) a PSM cell can withhold before fleeing the density,

maxPZ=4, representing the maximal density (number of cells in a neighborhood of size $2\varepsilon$) a progenitor cell can withhold before fleeing the density,

$\Delta t$=0.01, the time step of the scheme, representing 6 s of real biological development,

Tmax=60, representing 10 hr of real biological development,

$K_x$=0.43,

$K_y$=0.43,

$R^\star = 0.8$,

$\tilde{R} = 0.2$.

Case 1: high Sox2

To simulate the high Sox2 case in the mixed model, the only parameter we change is the noise in the equation of the ratio $R_i(t)$. In fact, instead of having a uniform variable in $[-1; 1]$, allowing cells to have an equal probability of differentiating to either PSM or neural cells, we shift the noise and insert a uniform variable in $[-0.92; 1.08]$. By doing so, cells have a higher probability to differentiate toward a neural ratio ($R_i(t)$=1).

Case 2: high Bra

To simulate the high Bra case in the mixed model, the only parameter we change is the noise in the equation of the ratio $R_i(t)$. In fact, instead of having a uniform variable in $[-1; 1]$, allowing cells to have an equal probability of differentiating to either PSM or neural cells, we shift the noise and insert a uniform variable in $[-1.08; 0.92]$. By doing so, cells have a higher probability to differentiate toward a PSM ratio ($R_i(t) = 0$).

## Figures

In the following paragraph, we present the details of the computations of the violin plots and the angle plots. We used this approach to validate the model and compare its results to those of the live imaging analysis.

We do the same computations for all three models.

## Cell tracking

To generate the violin plots, we save the cells' positions every 6 min (in real biological time, to correspond to the time points of the live imaging), this corresponds to 60 $\Delta t$ in simulation time with our choice of $\Delta t$. To compute the cell velocity in each tissue, we consider a region of size $1u \times 1.5u$ in the PZ and the NT, and $0.8u \times 2u$ in the PSM. At each time step, these regions are translated posteriorly, following the displacement of the most anterior point of the PZ.

Then, for each tissue, we compute the velocity of each trajectory for the cells in the considered regions, between the time point t and t+6 min, starting from t=0 to t=10 hr. We only consider the cells initially present in the regions and up to the time when they exit the regions.

To plot the distribution of the angles in each tissue for every cell type we consider each cell (in the regions previously drawn in each tissue) and compute the angle between the trajectory and the reference vector pointing downwards (in the direction of the elongation) of each trajectory, between the time point t and t+6 min. This angle will fall into one of 24 bins (dividing the 360° disk into 24 subdivisions). Finally, we compute the mean velocity in every bin. We then multiply each mean velocity, in each bin, by the proportion of trajectories in that bin, this gives the length of the arrow plotted in each bin.

## Tracking the shape of the progenitor zone

We upload the simulation generated by Matlab to the software ImageJ and manually track the most anterior and most posterior point of the PZ (per frame). This gives a track of the length of the PZ through time. For the width, as the PZ is moving and changing shapes, we choose to track the width of the mid-PZ throughout the simulation (tracking the mid-left and mid-right points). We then plot the rectangles hereby generated ($length \times width$) for initial time and final time.

We use the manual tracking of the most posterior point of the PZ to compute the elongation rate.

## Mean square displacement

To compute the MSD of progenitor cells we use the following formula:

$$MSD(t) = \frac{1}{N} \sum_{i=1}^{N} \left( (x_i(t) - x_i(0))^2 + (y_i(t) - y_i(0))^2 \right)$$

with N the number of cells considered. Here, we consider all the progenitor cells up to the time when they differentiate.

## Numerical scheme

We use the forward Euler method. The code was done using Matlab R2020a. The file contains three simulations:

- Simulation 1: Wild-type embryo (mixed model)
- Simulation 2: Case of high Sox2 and case of high Bra
- Simulation 3: Three models (random, mixed, and gradient)

