## [Decision Letter]

**Acceptance summary:**

Romanos and colleagues reveal extensive cell-to-cell heterogeneity in levels of *Sox2* and Bra in the region containing progenitors for neural and paraxial mesoderm. Gradually, this resolves such that cells with high Bra/low *Sox2* are in the mesoderm versus those with high *Sox2*/low Bra in the neurectoderm. Modulating *Sox2* and Bra bias the cells with high *Sox2* toward neurectoderm and high Bra toward mesoderm, with the latter being more motile. Finally, they mathematically model these behaviors. The study is nicely done and will be of broad interest to the readership of *eLife*.

**Decision letter after peer review:**

Thank you for submitting your article "Cell-to-cell heterogeneity in *Sox2* and Bra expression ratios guides progenitor destiny by controlling their motility" for consideration by *eLife*. Your article has been reviewed by 3 peer reviewers, and the evaluation has been overseen by a Reviewing Editor and Marianne Bronner as the Senior Editor. The following individual involved in review of your submission has agreed to reveal their identity: Valerie Wilson (Reviewer #2).

The reviewers have discussed the reviews with one another and the Reviewing Editor has drafted this decision to help you prepare a revised submission.

Essential revisions:

The reviewers raise several issues that need to be addressed, as detailed in the full reviews below. While the work nicely establishes correlations, we ask that you tone down arguments regarding causality. Specifically, we ask for better analysis of the existing data on effects of up- or down-regulating Sox2 and Bra, as well as improving the model to better fit the experimental data.

*Reviewer #1 (Recommendations for the authors):*

1. It would be important to describe better the experiments in which cell motility was measured. For instance, in the experiment shown in Figure 3 it is not clear where cells were electroporated. I assume that it was in the progress zone, but was this the case? If it was indeed the case and the image on Figure 3A is a representative example of an embryo in which motility was measured, I cannot see cells in the region of the neural tube, whereas there are many in the mesoderm, even at anterior levels. Does this introduce a bias to the results?

2. In the images shown in Supplemental Figure 2 it is hard to see the claimed "high degree of cell to cell heterogenicity" that is supposed to illustrate.

3. The authors state "overexpression of Bra leads to a marked reduction of the fluorescent signal in the PZ (1.17% {plus minus} 0.57) and to an increased signal in the PSM (33.04% {plus minus} 4.06) but has no effect on the NT signal (Figure 2 B, C, E)". However, in the image shown in Figure 2C there is a clear reduction of the number of cells in the PSM compared to the control and no obvious signals in the NT. A similar problem affects the experiment overexpressing *Sox2*. In particular, according to the description in the text "overexpression of *Sox2* drives exit of the cell from the PZ (1.16% {plus minus} 0.67) favoring their localization in the NT (75.40% {plus minus} 4.57) without significantly affecting proportions of cells in the PSM (Figure 2 B, D, E)". However, in the image shown in Figure 2D all the signal is in the NT, with no obvious labelled cell in the PSM. If these are representative images, I do not see that the description in the text matches the images. Also, how can this type of images lead to the quantification shown in panel E of this figure?

4. The number of labelled cells in the embryos after overexpression of *Sox2* or Bra is clearly much less than in the control. Does it mean that overexpression of *Sox2* or Bra blocks cell division or increase cell death?

5. In Figure 4, the image corresponding to the simulation after 10 hours shown in panel F is the same as the one shown in the control of panel J, but in this case representing 5 hours of simulation (at least according to the Figure legend). Can this be clarified?

*Reviewer #2 (Recommendations for the authors):*

Recent data from chimeras made with Bra mutant and wildtype cells suggests that Bra does not directly repress *Sox2* (doi 10.1016/j.devcel.2020.11.013), and in fact morpholino knockdown of Bra does not lead to a significant reduction in the level of *Sox2* (Figure S5H). The figure showing the effect of overexpressing Bra seems to show cells at either side of the midline (Figure 2C), ie not in the same pattern as the control vector (Figure 2B). Transverse views should be presented to be convincing that cells are in fact in the neural tube, as scored in Figure 2E, rather than in the medial part of the presomitic mesoderm. If it is the latter, this might explain the apparent tolerance of Bra epression in the neurectoerm. Furthermore, I would have expected from mouse data on Bra knockout cells in chimeras that cells would accumulate in the PZ. However Figure 2H, showing a Bra MO-treated embryo, shows accumulated cells in the posterior part of the neurectoderm- scored as neural tube. Is this in fact an abnormally-shaped PZ? Again sections may be helpful to convince the reader that these cells have really exited the PZ and become neurectoderm. Finally, the authors might be able to usefully comment on whether the ratio of *Sox2*/Bra is most important, or whether the absolute levels of each protein contribute to the outcome.

The authors only show limited data on the dorso-ventral organisation of the PZ (Figure 1 and S3). Clearly a part of the heterogeneity in Sox/Bra ratios correlates with the dorso-ventral axis, with *Sox2* dominating dorsally and Bra ventrally. This is how I would expect Sox and Bra to be expressed- they should be coexpressed in the PZ epithelium and *Sox2* should be downregulated as cells enter the paraxial mesoderm to either side. Likewise I would have expected to see dorsal to ventral movement, as well as mediolateral movement as cells exit the PZ to the paraxial mesoderm. The PZ cell tracks seem to indicate mediolateral movement (Figure 3B-corrected tracks) and so I can't quite understand how this is not more prominent on the radial plots in Figure 3D. This may need more explanation, as the rest of the manuscript, including the modelling, works on the assumption that cell motility is random in the PZ. Intuitively, I would have thought that if random motility is the only factor resulting in segregation of mesoderm/neurectoderm, raising *Sox2*/lowering Bra in cells would lead to nests of coherent low-motility cells rather than a coherent single neural tube flanked by paraxial mesoderm. Is this what is seen in the slightly branched neural tubes in Figure 4F and J? It doesn't seem to be the case after the experimental manipulations shown in Figure 2.

*Reviewer #3 (Recommendations for the authors):*

– My main concern is whether cell motility of NMPs is upstream or downstream of the fate choice. Establishing the timing of cell fate commitment with additional markers cell transplantation experiments, or single-cell experiments could be used to clarify this.

– Additional evidence that supports a role of *Sox2*/Bra as regulators of cell migration would strengthen the study – these could include migration assays, use of EMT/migration markers, precise manipulation of *Sox2*/Bra levels.

– Measurements of cell motility in the PZ and mesoderm (Figure 3) suggest that NMPs move as much as cells in the mesoderm. Why does the model consider that NMPs move less than the mesoderm? The authors should clarify this.

[Editors' note: further revisions were suggested prior to acceptance, as described below.]

Thank you for submitting your article "Cell-to-cell heterogeneity in *Sox2* and Bra expression guides progenitor motility and destiny" for consideration by *eLife*. Your article has been reviewed by 3 peer reviewers, and the evaluation has been overseen by a Reviewing Editor and Marianne Bronner as the Senior Editor. The following individuals involved in review of your submission have agreed to reveal their identity: Moises Mallo (Reviewer #1); Valerie Wilson (Reviewer #2).

I am happy to say the reviewers were very pleased with your revisions and only ask for a few clarifications as detailed below.

Essential revisions:

1. The sections in Figure 2—figure supplement 3 illustrate well that the effects on the differentiated cells of Pcig-*Sox2* versus Bra-MO are quite similar (decreased mesoderm contribution). However the authors don’t address the question of whether an accumulation of cells in Bra-MO electroporated embryo PZ (specifically in the CNH) has been overlooked. To my eye, it seems that this is visible in section F” of. Comparing the distribution of electroporated cells in the embryos shown, it seems that F” (cells expressing Bra-MO) shows a high proportion of electroporated cells in an abnormally thickened posterior-ventral neural tube (the position looks comparable to that of the section in B”). This is a more posterior location that C”, showing cells overexpressing Pcig-*Sox2*. It seems to support the idea that knocking down Bra leads to accumulation of cells in the (CNH) of the progenitor zone, as seen before in mouse studies, so that the effect on the progenitors of reducing Bra is not the same as increasing *Sox2*. I think it would be worth including this in the discussion, if these sections are representative of all embryos of this type.

2. Related to this, it would be helpful if the position of the end of the notochord could be marked in the wholemount picture, to orient the reader.

3. The statement that ‘the proportion of signal quantified within the neural tube on whole mount (Figure 2E) is probably over-estimated’ (line 196-7) is slightly problematic as it raises a doubt in the reader’s mind about the accuracy of the quantitation. To address this completely, it would be necessary to add data showing the correlation between automated counts and a sample of corresponding manual counts. Alternatively, providing an explanation or a qualifying sentence that says why this is irrelevant to the main argument might suffice.

4. In the response to reviewers, the authors show data supporting the idea that there are no major effects on cell death or cell cycle. It would help the authors’ case to include this in the supplementary data.

---

## [Author Response]

Essential revisions:The reviewers raise several issues that need to be addressed, as detailed in the full reviews below. While the work nicely establishes correlations, we ask that you tone down arguments regarding causality. Specifically, we ask for better analysis of the existing data on effects of up- or down-regulating Sox2 and Bra, as well as improving the model to better fit the experimental data.

Altogether, we think that the editor and reviewers’ comments were very helpful in pointing out weak points in the previous version of the manuscript and we want to thank them for that. Having addressed their comments, we are submitting a revised version of the manuscript which is more precise and accurate, and therefore, we believe that this new version is now suitable for publication in eLife.

Concerning the claims about causality that we made in the previous version of the manuscript, we agree with reviewers and editor that we do not strictly demonstrate the requirement of spatial cell-to-cell heterogeneity (heterogeneity in the sense of a “random-like” pattern of Sox2/Bra ratios) in NMP maintenance. We also agree that, although our data clearly show a role of Sox2 and Bra in controlling progenitor motility, we cannot ascertain that Sox2 and Bra direct progenitor tissue distribution solely by controlling their motility. We have thus toned down our arguments in the revised version of our manuscript. In particular, parts of the text stating that heterogeneity is necessary for NMPs maintenance have been removed and conclusions that the effects of Sox2 and Bra on progenitor distribution are going through the control of their motile behaviors have been tempered (detailed in response to reviewers).

In the revised version of our manuscript, we also provide better illustrations of the Sox2/Bra phenotypes (more representative images have been included in Figure 2) but also additional data we obtained from the analysis of embryonic transverse sections for all functional experimental conditions (Figure 2-figure Supplement 3). We also approached the question of the effects of Sox2 and Bra on cell fate maturation in the PZ and provide some evidence that progenitors do not yet express differentiation markers as they acquire specific motile properties in response to Sox2 or Bra overexpression (Figure 4-figure Supplement 2). We also analyzed cell proliferation and cell death and did not found differences in response to Sox2 and Bra overexpression or downregulation (detailed in response to reviewer 1). We as well included new data involving a Sox2 reporter gene that support heterogeneity of cell motile behaviors within the PZ (Figure 3-figure Supplement 1B and Video 2).

Furthermore, we have improved the modelling section of the manuscript by developing a new model that better fits the biological data (new Figure 5). To evaluate the influence of the different types of heterogeneity distributions (random versus gradient), this model has been compared to the two models we proposed in the previous version of our manuscript (new Figure 6). This allows us to propose some roles of “random-like” heterogeneity in morphogenesis. To facilitate the flow of the manuscript, we have moved the modeling section to the end of the manuscript.

Reviewer #1 (Recommendations for the authors):1. It would be important to describe better the experiments in which cell motility was measured. For instance, in the experiment shown in Figure 3 it is not clear where cells were electroporated. I assume that it was in the progress zone, but was this the case? If it was indeed the case and the image on Figure 3A is a representative example of an embryo in which motility was measured, I cannot see cells in the region of the neural tube, whereas there are many in the mesoderm, even at anterior levels. Does this introduce a bias to the results?

Electroporation were performed at stage HH5, targeting cells of the anterior primitive streak and of the epiblast that contain both neural and mesodermal progenitors. This is described in results ( l.163-168) and in the section Material and Methods (electroporation paragraph, l. 496). Reference 60, a reference which is particularly relevant to describe this technique, is cited in the section Material and Methods. Details of motility measurements are presented in the supplemental methods. Figure 3A has been revised to give a better representation of results.

2. In the images shown in Supplemental Figure 2 it is hard to see the claimed "high degree of cell to cell heterogenicity" that is supposed to illustrate.

We have added some arrows and changed the Supplemental Figure 2 legend to help readers better see the heterogeneity.

3. The authors state "overexpression of Bra leads to a marked reduction of the fluorescent signal in the PZ (1.17% {plus minus} 0.57) and to an increased signal in the PSM (33.04% {plus minus} 4.06) but has no effect on the NT signal (Figure 2 B, C, E)". However, in the image shown in Figure 2C there is a clear reduction of the number of cells in the PSM compared to the control and no obvious signals in the NT. A similar problem affects the experiment overexpressing Sox2. In particular, according to the description in the text "overexpression of Sox2 drives exit of the cell from the PZ (1.16% {plus minus} 0.67) favoring their localization in the NT (75.40% {plus minus} 4.57) without significantly affecting proportions of cells in the PSM (Figure 2 B, D, E)". However, in the image shown in Figure 2D all the signal is in the NT, with no obvious labelled cell in the PSM. If these are representative images, I do not see that the description in the text matches the images. Also, how can this type of images lead to the quantification shown in panel E of this figure?

We have changed the images to be more representative of the quantified results. The methods of the quantification are detailed in the text (L.172) and in the Image acquisition and processing section of the Material and Methods (L.526-532). To back up or relativize our claims, we have also proceeded to transversal sectioning of representative embryos (L. 192, Figure 2—figure supplement 3).

4. The number of labelled cells in the embryos after overexpression of Sox2 or Bra is clearly much less than in the control. Does it mean that overexpression of Sox2 or Bra blocks cell division or increase cell death?

Again, we apologize for images of Figure 2 in the previous version of the manuscript being not representative of the quantifications. We have changed them to more representative images. As we tracked progenitors by live imaging, we did not noticed massive cell death after overexpression of *Sox2* and Bra. However, to make sure and to answer the reviewer’s comment, we analyzed cell death (Caspase-3 staining) and proliferation (EdU pulse of 1hour) on transfected cells 7hrs after overexpression. Here again, we did not find striking differences between controls (pcig) and cells misexpressing *Sox2* or Bra (pcig-*Sox2*, pcig-bra). We chose to display these results to the reviewer:

5. In Figure 4, the image corresponding to the simulation after 10 hours shown in panel F is the same as the one shown in the control of panel J, but in this case representing 5 hours of simulation (at least according to the Figure legend). Can this be clarified?

We thank the reviewer for drawing our attention to this issue. Modification of the Figure legend has been done.

Reviewer #2 (Recommendations for the authors):Recent data from chimeras made with Bra mutant and wildtype cells suggests that Bra does not directly repress Sox2 (doi 10.1016/j.devcel.2020.11.013), and in fact morpholino knockdown of Bra does not lead to a significant reduction in the level of Sox2 (Figure S5H). The figure showing the effect of overexpressing Bra seems to show cells at either side of the midline (Figure 2C), ie not in the same pattern as the control vector (Figure 2B). Transverse views should be presented to be convincing that cells are in fact in the neural tube, as scored in Figure 2E, rather than in the medial part of the presomitic mesoderm. If it is the latter, this might explain the apparent tolerance of Bra epression in the neurectoerm. Furthermore, I would have expected from mouse data on Bra knockout cells in chimeras that cells would accumulate in the PZ. However Figure 2H, showing a Bra MO-treated embryo, shows accumulated cells in the posterior part of the neurectoderm- scored as neural tube. Is this in fact an abnormally-shaped PZ? Again sections may be helpful to convince the reader that these cells have really exited the PZ and become neurectoderm. Finally, the authors might be able to usefully comment on whether the ratio of Sox2/Bra is most important, or whether the absolute levels of each protein contribute to the outcome.

We agree that Bra Knockdown at 7h does not lead to a significant reduction of *Sox2* questioning a strict and direct inhibitory relationship between the two factors. This aspect is discussed in the new version of our manuscript (L.388-391).

We now provide transverse sections of embryos misexpressing *Sox2* or Bra (Figure 2 figure supplement 3). These data show that most of Bra overexpressing cells were excluded from the neural tube indicating that the proportion of signal quantified within the neural tube on whole mount (Figure 2E) is most likely over-estimated stated in the new manuscript in (L.192-196). These cells are mainly located in the most medial part of the PSM, which is quite different from what we observed for control cells. We have not yet found a good explanation for this particular distribution (a link might be done with the fact that these cells maintain high level of Bra expression). However, we also found few Bra overexpressing cells located in the neural tube in our sections (Figure 2 figure supplement 3B’’). As indicated in the Discussion section of the manuscript, our interpretation for the presence of these cells in the neural tube is that they might correspond to progenitors already committed to the neural fate at the time of transfection. These experiments also allowed confirming that Bra-depleted cells (Bra MO) are clearly located in a correctly formed neural tube (Figure 2 figure supplement 3F’, F’’), thus eliminating the possibility that these cells remains an abnormally shaped PZ.

Solving the question of the contribution of either *Sox2*/Bra values or absolute levels of the two proteins would undoubtedly be of great interest (discussed L.391) but very challenging, in particular because changing the level of one of the two proteins has impact on the level of the other (at least in three of our experimental conditions). We are not thus in a good position to investigate this question within a reasonable time.

The authors only show limited data on the dorso-ventral organisation of the PZ (Figure 1 and S3). Clearly a part of the heterogeneity in Sox/Bra ratios correlates with the dorso-ventral axis, with Sox2 dominating dorsally and Bra ventrally. This is how I would expect Sox and Bra to be expressed- they should be coexpressed in the PZ epithelium and Sox2 should be downregulated as cells enter the paraxial mesoderm to either side. Likewise I would have expected to see dorsal to ventral movement, as well as mediolateral movement as cells exit the PZ to the paraxial mesoderm. The PZ cell tracks seem to indicate mediolateral movement (Figure 3B-corrected tracks) and so I can't quite understand how this is not more prominent on the radial plots in Figure 3D. This may need more explanation, as the rest of the manuscript, including the modelling, works on the assumption that cell motility is random in the PZ. Intuitively, I would have thought that if random motility is the only factor resulting in segregation of mesoderm/neurectoderm, raising Sox2/lowering Bra in cells would lead to nests of coherent low-motility cells rather than a coherent single neural tube flanked by paraxial mesoderm. Is this what is seen in the slightly branched neural tubes in Figure 4F and J? It doesn't seem to be the case after the experimental manipulations shown in Figure 2.

A parasagittal optical section of the posterior tissue has been added (Figure 1 figure supplement 3) to better document the DV organization.

It is difficult with the low Z resolution of our live imaging technique to document DV movements but we agree that given to the *Sox2* and Bra expression patterns, differential DV movements likely take place. The PZ tracks showed on the previous figure 3B, which were done for few cell tracks (around 30) coming from one embryo, were not representative and gave the wrong impression of lateral movements. Modification of the Figure 3 has been done and a new illustration, which is more representative, is provided in the new Figure 3. The plot of the angles (radial plot), which represents more tracks (>500) coming from 7 distinct embryos, are more representative do not show this lateral bias.

In our model, this is not motility alone but also non-mixing and adhesion (*Sox2* favoring adhesion) properties that allow the segregation of cell populations, we explained it better in the new version of the manuscript (L. 295-300); We also tested and verified the requirements of these two parameters in our new model (Figure 5F). Therefore, the fact that *Sox2* favors adhesion leads to more sticky cells in the *Sox2* overexpression conditions of our model, possibly contributing to the formation of branched neural tubes observed in the simulation. Interestingly, we do see *Sox2* OE cells sticking together outside of the NT in our experiments too. This is not captured by our imaging in Figure 2, however to answer reviewer comment we decided to display such phenomenon on confocal data, see Author response image 1. These data argue in favor of a role of cell adhesion downstream of *Sox2*, a hypothesis that we discuss in the article (L. 457) and that we will pursue in our future works.

Reviewer #3 (Recommendations for the authors):– My main concern is whether cell motility of NMPs is upstream or downstream of the fate choice. Establishing the timing of cell fate commitment with additional markers cell transplantation experiments, or single-cell experiments could be used to clarify this.

This is a very interesting point, which is difficult to address because cells that change their motility rapidly exit the PZ. We decided to study the expression of Pax6 for *Sox2* OE and Msgn1 for Bra OE 7h after electroporation, a short window of time when we already see motility changes and when the progenitors are still in the PZ. We did not observe drastic changes in their neural and mesodermal fates (Figure 4 figure supplement 2). These data suggest that *Sox2* and Bra can affect motility before affecting the expression of fate makers Pax6 and Msgn1.

– Additional evidence that supports a role of Sox2/Bra as regulators of cell migration would strengthen the study – these could include migration assays, use of EMT/migration markers, precise manipulation of Sox2/Bra levels.

We have recent evidence that *Sox2* and Bra regulates the expression of some adhesion molecules in the PZ . However, these data are very preliminary and will be at the basis of future studies in the lab.

– Measurements of cell motility in the PZ and mesoderm (Figure 3) suggest that NMPs move as much as cells in the mesoderm. Why does the model consider that NMPs move less than the mesoderm? The authors should clarify this.

We probably underestimated slower cells at the dorsal side of the embryo as we are imaging from ventral side of the embryo, therefore we thought that we should take into account these unseen cells in the model. To visualize better these slower cells, we used a *Sox2* reporter gene in the new version of the manuscript. We were able to track GFP positive cells and to confirm that these cells migrate slower than the GFP negative cells reinforcing the fact that there are indeed slower cells in the PZ; we have included these results in the main text L225-232 and in (Figure 3 figure supplement 1B).

[Editors' note: further revisions were suggested prior to acceptance, as described below.]

Essential revisions:1. The sections in Figure 2—figure supplement 3 illustrate well that the effects on the differentiated cells of Pcig-Sox2 versus Bra-MO are quite similar (decreased mesoderm contribution). However the authors don't address the question of whether an accumulation of cells in Bra-MO electroporated embryo PZ (specifically in the CNH) has been overlooked. To my eye, it seems that this is visible in section F" of. Comparing the distribution of electroporated cells in the embryos shown, it seems that F" (cells expressing Bra-MO) shows a high proportion of electroporated cells in an abnormally thickened posterior-ventral neural tube (the position looks comparable to that of the section in B"). This is a more posterior location that C", showing cells overexpressing Pcig-Sox2. It seems to support the idea that knocking down Bra leads to accumulation of cells in the (CNH) of the progenitor zone, as seen before in mouse studies, so that the effect on the progenitors of reducing Bra is not the same as increasing Sox2. I think it would be worth including this in the discussion, if these sections are representative of all embryos of this type.

We agree that there is a thickening of the ventral part of the neural tube on section F’’ (Bra Mo), this is something we often observed on different embryos. Our interpretation is that this thickening is due to Bra-Mo electroporated progenitors which have leaved the PZ and are integrating into the neural tube. At the level of this section, the notochord and the neural structure are indeed separated (the notochord being the round, GFP negative, structure located ventrally on the section). In the section F’’’, which has been done at the level of the PZ, few fluorescent cells can be seen in the ventral zone. We agree that it does not seem to be the case in the *Sox2* OE conditions at a comparable antero-posterior level (C’’’). This difference is however difficult to interpret as resulting from distinct effects of *Sox2* OE or Bra KD since differences might simply result from distinct efficacies of the methods we used to impair protein levels (electroporation of MO versus expression vector). What clearly appears on these sections is that the number of Bra Mo left in the PZ (F’’’) is lower than that observed in the control Mo section done at an equivalent antero-posterior level (D’’’), confirming the fact that in Bra-Mo conditions a greater proportion of electroporated cells have left the PZ compared to control conditions (quantified in Figure 2J). The main purpose of this supplemental figure was to verify the cellular localizations, either in the PZ, the PSM and the NT. We did not aim at quantifying fluorescence in the CNH (a structure which is described to be located in the tail bud, hence at later stages of development than the ones we used in our experiments). Analysis of the CNH ( as the anterior most part of the PZ located just posterior to the tip of the notochord) would probably have been better-assessed at later stages using sagittal sections in order to make sure to visualize this particular region along the A/P axis. Our results indeed confirmed that most Bra-Mo cells have left the PZ to locate into the NT; however, we don’t want to rule out the possibility that some of the Bra-Mo electroporated cells which are still in the PZ at this stage are stuck in a region that will form the CNH later on. We have included this particular aspect in the discussion (L.409):

“Down-regulation or loss of Bra expression have been associated with retention of cells in the progenitor regions in mouse embryo studies, particularly in the tail bud at the level of the CNH (Chordo-Neural Hinge) (19,20). Although we observed a clear decrease in the number of PZ cells in Bra-Mo compared to control conditions, it is possible that some of the Bra-Mo cells that are remaining in the PZ indeed reside in a region that will contribute to the CNH.”

Finally, to clarify our interpretations for readers, we have also indicated that only panels (‘’’) show the ZP in the legend of Figure 2-Supplement 4 (former Supplement 3) and positioned the posterior limit of the notochord.

2. Related to this, it would be helpful if the position of the end of the notochord could be marked in the wholemount picture, to orient the reader.

We have added arrowheads at the position of the posterior end of the notochord on Figure2- Supplement 4.

3. The statement that 'the proportion of signal quantified within the neural tube on whole mount (Figure 2E) is probably over-estimated' (line 196-7) is slightly problematic as it raises a doubt in the reader's mind about the accuracy of the quantitation. To address this completely, it would be necessary to add data showing the correlation between automated counts and a sample of corresponding manual counts. Alternatively, providing an explanation or a qualifying sentence that says why this is irrelevant to the main argument might suffice.

After analyzing transverse sections of multiple embryos, we are convinced that despite the small number of mesoderm signal that could have been counted as “neural” because it is located ventrally to the neural tube in the Bra OE, quantifications done in 2D are precise enough to support our claims. We apologize that our statement could have raise doubts in the reader’s mind; we have changed it and provided an explanation that these small differences are actually irrelevant to our main conclusions (L.197):

“It must be noticed that, transverse sections showed the presence of a large proportion of Bra-overexpressing cells located in the medial part of the paraxial mesoderm, very close to the neural tube, while only few Bra-overexpressing cells were indeed located into the neural tube. We thus can’t exclude the possibility that, due to this particular cell distribution, the neural tube signal quantified on whole mount embryos might have been slightly overestimated (Figure 2E). Even if it were the case, it does not question our main conclusions that Bra overexpression, in comparison to control conditions, favors the exit of progenitors from ZP and their subsequent localization into the paraxial mesoderm.”

4. In the response to reviewers, the authors show data supporting the idea that there are no major effects on cell death or cell cycle. It would help the authors' case to include this in the supplementary data.

The data have been added as Figure 2-supplement Figure 3, text has been added L.185 of the manuscript:

“To verify that the differences in fluorescence distributions we observed did not result from distinct apoptotic or proliferation rates, we quantified these parameters 7 hrs after electroporation. Our data showed no major changes between the different experimental conditions, validating that protein misregulations indeed act by influencing the distribution of cells in the different tissues (Figure 2 —figure supplement 3).”